

# Detection and reconstruction of rock glaciers kinematic over 24 years (2000-2024) from Landsat imagery

Diego Cusicanqui[1], Pascal Lacroix[1], Xavier Bodin[2], Benjamin Aubrey Robson[3], Andreas Kääb[4] and Shelley MacDonell[5,6]

[1]Institut des Sciences de la Terre (ISTerre) CNES, CNRS, IRD, Univ. Grenoble Alpes, Grenoble, 38000, France
[2]Laboratoire EDYTEM, Univ. Savoie Mont-Blanc, Le Bourget du Lac, 73376, France
[3]Department of Earth Science, University of Bergen, Bergen, Norway
[4]Department of Geosciences, University of Oslo, Oslo, 0316, Norway
[5]Centro de Estudios Avanzados en Zonas Áridas (CEAZA), La Serena, Chile
[6]Waterways Centre, University of Canterbury and Lincoln University, Christchurch, New Zealand

*Correspondence to*: Diego Cusicanqui (diego.cusicanqui@univ-grenoble-alpes.fr)

**Abstract.** The importance of monitoring rock glacier dynamics is now widely acknowledged within the scientific community following the designation of rock glacier velocity as a parameter of the Essential Climatic Variable permafrost. However, the representation of long-term spatio-temporal patterns of rock glaciers velocity at regional scale remains challenging due to the unavailability of high-resolution remote sensing datasets. This study presents a robust methodological approach based on the redundancy of information, joint with the inversion of surface displacement time series and the automatic detection of persistent moving areas (PMA) applied to rock glacier monitoring, using annual open-access, medium-resolution Landsat 7/8 optical imagery. This methodology enables the detection, quantification and analysis of surface kinematics of 382 gravitational slope movements over a 24-years, of which 153 corresponds to rock glaciers. This is the first time that Landsat images were used to quantify rock glacier displacements and derived velocities. The results demonstrate an average velocity of 0.37 ± 0.07 m y-1 overall 24-year for all rock glaciers, with some exceptions where large rock glaciers and debris frozen landform exhibit surface velocities exceeding 2 m y-1. The results of this study shows a good agreement with high-resolution imagery and recent GNSS measurements. L7/8 imagery tends to underestimate surface velocity by approximately 10-20%. The intrinsic limitations of Landsat imagery make it challenging to interpret annual velocity variations. Notwithstanding, decadal velocity changes can be depicted for the fastest and largest rock glaciers, revealing 10% of the accelerations in one decade. Our study suggests a correlation between surface velocity and local topographic parameters (orientation, slope, elevation) as possible controlling factors. In conclusion, this study demonstrates the feasibility of using medium-resolution optical imagery, providing an alternative to InSAR, for monitoring rock glacier kinematics anywhere over the World.



## 1 Introduction

Historically, the state of the cryosphere has been measured using specific variables defined by Global Climate Observing System (GCOS) (GCOS, 1995) such as mass balance for glaciers, snow covered variability for snow and soil temperature for permafrost. Among those variables, glacier mass balance and snow cover variations are relatively well known at global scale (Hugonnet et al., 2021; Notarnicola, 2020) compared to changes in mountain permafrost, which are still very incompletely
monitored (Bolch et al., 2020).

Permafrost is an important component of the cryosphere occurring at high latitudes (i.e. polar regions) and high elevations (i.e. mountainous areas). As permafrost —ground material remaining at or below 0°C for at least two consecutive years— is a thermal phenomenon, it is thus sensitive to changes in climate forcing (Hock et al., 2019). The evidence of mountain permafrost warming relies on very few direct borehole observations (Noetzli et al., 2019), distributed mostly in the western
Alps and Alaska. Thus, several mountain ranges are lacking sufficient permafrost monitoring data to analyse changes. Mountain permafrost degradation is indicated by the increase in ground temperatures and active layer thickness (Etzelmüller et al., 2020), the increase of liquid water content within the frozen terrain (Cicoira et al., 2019), as well as ground-ice melt (Cusicanqui et al., 2021; Haberkorn et al., 2021). However, warming affects mountain permafrost differently according to the type of terrain, reflecting in particular the influence of  snow cover on the ground. While a rather steady warming trend
can be observed in steep rock slopes (Magnin et al., 2024), where snow is poorly present, permafrost temperatures in loose rock formations such as rock glaciers show pronounced inter-annual variations, mostly due to the variable insulating effect of snow (Thibert & Bodin, 2022; Kellerer-Pirklbauer et al., 2024).

In the present paper, we follow the definitions proposed by the IPA Rock Glaciers Inventory and Kinematics (RGIK) action group, which states that rock glaciers can be defined as "debris landforms generated by the former or current creep of frozen
ground (permafrost), detectable in the landscape with the following morphologies: front, lateral margins and optionally ridge-and-furrow surface topography" (Berthling, 2011; RGIK, 2023). Given the complexity of measuring permafrost warming, rock glacier velocity has been recently proposed and accepted by the GCOS to be a complement of the Essential Climatic Variable ECV-permafrost (Hu et al., 2023). Indeed, the thermally-dependent creep of ice-rich frozen ground is inherently sensitive to climatic conditions and it is able to change over different timescales (Delaloye et al., 2010; Kääb et
al., 2007; Sorg et al., 2015). While the variation of creep velocities at inter-annual, seasonal rhythms and over shorter periods reflects mainly the influence of weather (Kenner et al., 2017; Wirz et al., 2016), long-term patterns (e.g. decadal to pluri-decadal scales) relate primarily to mean annual air or ground temperatures (Pellet et al., 2022, Kellerer-Pirklbauer et al., 2024). In arid zones, such as the semiarid Andes, rock glaciers have been considered as long-term storage of water (Azocar and Brenning, 2010; Trombotto et al., 1999), with the potential to enhance surface runoff within the active layer (Burger et
al., 1999). Although rock glaciers are much smaller than most glaciers, they could contain a considerable amount of water due to their widespread distribution (Brening, 2005). More recent estimations (e.g. Halla et al., 2021; Navarro et al., 2023; de





Pasquale et al., 2022) geophysical studies highlight the high ice and water content stored in rock glaciers. However, their significance at regional scales still has large uncertainties (Arenson et al., 2022, Wagner et al., 2021).

Since the early 2000s, there has been a growing interest from the international community in the monitoring of rock glacier velocity. Indeed, observations show that rock glacier velocities often exhibit similar interannual to longer term trends at regional scale (Kellerer-Pirklbauer & Kaufmann, 2021; Marcer et al., 2021; Pellet et al., 2022), which strongly depends on local ground temperature changes (Noetzli et al., 2019). The velocity of rock glaciers is controlled by the intrinsic characteristics of the landform, in particular its internal structure (ice / debris proportions, thickness) and the topography (slope of the bed), but it is also influenced substantially by a external climatically-driven factors such as ground temperature and and advection, infiltration, or internal production of water (Jansen and Hergarten, 2006; Cicoira et al., 2019a; Kenner et al., 2020). Thus, magnitude and variability of the velocity can thus give indications on the current state and possible ongoing changes in the characteristics of the permafrost body. The monitoring of changes in rock glacier velocity thus provides information about the impact of climate change on mountain permafrost kinematics and, indirectly, on its thermal state. Given the current warming context, creep speed of rock glaciers in cold mountains is expected to increase with ground temperature (Arenson et al., 2015; Kääb et al., 2007; Müller et al., 2016).

Quantifying rock glacier velocity over regional scales has been best done using satellite radar interferometry (InSAR) data. This method enables detection of slow slope movement (i.e. rock glacier motion) in the Line of Sight (LOS) of the satellite, over large regions and hundreds of individual landforms (Hu et al., 2023). This approach has been used to map rock glacier motion around the world (Bertone et al., 2022). This data source has served as a base for classifying movements rates of orders of magnitude (cm/d, cm/month, dm/month, cm/yr, etc.), standardised within the (RGIK) group (RGIK, 2023). However, even if this technique is well suited for rock glacier mapping (Barboux et al., 2014) satellite radar interferometry is mainly useful for relatively slow rock glacier speeds, with maximum speeds about 1–1.5 m yr$^{-1}$. Beyond this threshold, InSAR signals become geometrically decorrelated and thus uninterpretable (Villarroel et al., 2018). In addition, InSAR data with high temporal repetition is only available since the early 21st-century (Strozzi et al., 2020). This means that no climatic timescales (i.e. decennial timescales) for rock glaciers could be obtained with this source of data.

On the other hand, optical imagery offers a more robust alternative where set tracking techniques can be applied to repeat imagery to derive surface displacements, both to contemporary imagery and historical datasets, allowing rock glacier velocities to be investigated over longer time-scales (Cusicanqui et al., 2021; Kääb et al., 2021; Kaufmann et al., 2021). This technique is not well suitable for slow movements due to its low signal-to-noise ratio (unless very high spatial resolution allows tracking the movement), but rather well suitable for medium to large movements beyond 1 - 1.5 m yr$^{-1}$ (Hartl, et al., 2021; Marcer et al., 2021). However, to date such techniques have been exemplified on rock glaciers using high resolution optical imagery (< 5 m). Very often, using airborne imagery that is not easily accessible can be prohibitively expensive for larger regions or for more extensive time series. As a consequence, few periglacial areas have been extensively investigated using feature tracking, with most of the studies restricted to the European Alps (Cusicanqui et al., 2021; Eriksen et al., 2018;



Hartl et al., 2016; Kellerer-Pirklbauer and Kaufmann, 2012) and some isolated regions in the Andes i.e. Tapado rock glacier (Vivero et al., 2021) and in northern Tien Shan (Kääb et al., 2021).

Medium resolution imagery (KH-9, Landsat-7, Landsat-8, SPOT 1-4,...) offers a continuous dataset for monitoring slow-moving landforms since the 1980's. Recent progress in time-series processing has allowed the development of methods for both detecting and monitoring slow-moving landslides using medium-resolution imagery over the last 40-50 years

(Bontemps et al., 2018; Lacroix et al., 2020a). These methods have never been used for rock glaciers because of (1) the rather slow motion of rock-glaciers overall ($\sim$ 1 m yr$^{-1}$), and (2) the difficulty of the processing caused by the presence of snow and shadows in mountain topography. Here, we demonstrate the applicability of the free and open-access, global, medium-resolution satellite datasets Landsat 7/8 (called hereafter L7/8) to characterise rock glacier displacements and velocity for the late 20th century in a region of the semiarid Andes (both on Chile and Argentina). We further validate our

results ate regional scale using Sentinel-1 wrapped interferograms, and a more local scale with very high resolution datasets e.g. Geoeye, Pléiades, airborne (called hereafter VHR) on the Tapado complex area and recent Global navigation satellite system (GNSS) measurements.

## 2 Study area and previous work

Our study area is located within the Coquimbo and San Juan provinces, in the semiarid Andes of Chile and Argentina

(between 29°20'S and 31°15'S latitude; Fig. 1). It is a ~45x45 km² surface, with altitudes ranging between 3,000 and 6,300 m a.s.l. According to the national inventories of Chile (DGA, 2022) and Argentina (IANIGLA, 2018), both obtained from geomorphological interpretation of optical satellite imagery, the area has a relatively high number of rock glaciers with 80 located on the Chilean side and 235 on the Argentinian side (Fig. 1).

Briefly, the regional climate is characterised by semiarid conditions, influenced mainly by the subtropical South Pacific

anticyclone (Montecinos & Aceituno, 2003). The rapid elevation changes in topography from coastal position to the high elevation of Andes mountain range (~6,000 m.a.s.l.) have a strong influence on the general atmospheric circulation, notably differentiating the eastern and western climatic systems (Kalthoff et al., 2002). Schauwecker (2022) shows that the precipitation coming from the humid Pacific masses occurs almost exclusively as snowfall and mostly concentrated in the austral winter, between May and August. Year-to-year precipitation varies notably in accordance with the El Niño or ENSO

(El Niño Southern Oscillation) phenomenon with above (below) -average precipitation during El Niño (La Niña) events (Masiokas et al., 2006, 2010) with recent deficits in precipitations between 20-40% (Garreaud et al., 2020). Recent meteorological records on three Automatic Weather Stations (AWS) show mean annual precipitation of ~170 mm in the last decade (CEAZA, 2023).

Recent studies of the air temperature have shown a trend of 0.2°C per decade in the central Andes, closely influencing the

decrease in snowfalls (Poblete & Minetti, 2017; Réveillet et al., 2020). According to global permafrost distribution models



(Gruber, 2012; Obu, 2021) and a local one (Azócar et al., 2017), heterogeneous/discontinuous permafrost is present between 3,900 - 4,500 m a.s.l., becoming more prevalent above 4,500 m a.s.l. (Fig. 1).

The study of mountain permafrost in this particular region of the semiarid Andes has received attention during the last decades, because of the high density and large extension of rock glaciers (Janke et al., 2015), as well as for their high public understanding hydrological significance (MacDonell et al., 2022). In this context, several rock glacier inventories have been carried out along the Chilean (e.g. DGA, 2022) and Argentinian (e.g. IANIGLA, 2018) Andes. Several detailed/local geomorphological investigations (Monnier & Kinnard, 2015, 2016, Halla et al., 2021; Navarro et al., 2023; de Pasquale et al., 2022) have been carried out.These studies highlight the complex interaction between remnants of glaciers, debris covered glaciers and rock glaciers (Navarro et al., 2023a; Robson et al., 2021) as well as the importance of rock glaciers as water storage resources (MacDonell et al., 2022; Schaffer et al., 2019; Schaffer and MacDonell, 2022), and change of features through time (Robson et al., 2022).

Despite this state-of-the-art, limited overview of the rock glacier velocity exists, while historical trends of velocity largely remain unknown. For instance, Villarroel et al. (2018) provided a recent kinematic inventory of the Argentinean Andes, between 30.5°S and 33.5°S, identifying ~2100 active rock glaciers based on InSAR. On the other hand, Blöthe et al (2021) provided a regional assessment in the "Cordon del Plata" range (~300 km south of our study area), quantifying velocity fields of 244 rock glaciers between 2012 and 2015 using offset tracking optical imagery. Most of the rock glacier monitoring is focused on single rock glaciers i.e. Tapado complex in the Chilean (Vivero et al., 2021) and Dos Lenguas (Halla et al., 2021; Strozzi et al., 2020) in the Argentinian Andes. In this sense, a historical perspective is still lacking in the region. The majority of studies that have assessed rock glacier velocities in the area have utilised contemporary remote sensing datasets (last 10 years). Vivero et al., (2021) provides the longest surface velocity time series covering almost 70 years since the 1950's. Finally, this region was chosen due to good coverage of reference datasets, namely VHR satellite imagery and in situ GNSS measurements on the Tapado rock glacier (DGA, 2010), that could be used for validation.







**Figure 1:** Location of the study area in the semiarid Andes (between 29°20'S and 31°15'S latitude). Red square in the inner map shows the footprint of the Landsat scenes used in this study. Within the main map, black dots correspond to rock glacier inventory for Chile (DGA, 2022) and Argentina (IANIGLA, 2018). The orange-purple colorbar represents the Permafrost Favorability Index (PFI) from (Gruber, 2012). A comparison with the more recent PFI from Obu (2021) is shown in Fig. S1. Background map corresponds to © OpenTopoMap.

## 3 Data

Three different remote sensing datasets were used in this study: (1) L7/8 images, (2) VHR images from airborne platforms and satellites, used to validate the L7/8 products temporally, and (3) InSAR images produced from Sentinel-1 SAR satellites, used to validate the L7/8 products spatially. We also used GNSS data acquired on one specific rock glacier for the kinematic validation.



## 3.1 L7/8 dataset

The L7/8 dataset comprises freely and openly available 8-band multispectral orthorectified satellite images spanning the
period from 2000 to 2024 (Fig. 2a and b). However, a data gap existed on the L7/8 dataset between 2004 and 2013, due to
the failure of the Scan Line Corrector on the satellite (Markham et al., 2004). Here, we used only the L7/8 panchromatic
band (B8) with the highest pixel resolution (i.e. 15 m). All images correspond to path and row 233 / 081 and have been
cropped within a common grid (3001x3001 pixels) corresponding to a surface area of 45x45 km². One image per year was
chosen visually during the clear months of summer (January to April), with the goal of obtaining the least snow and cloud
cover possible (Table S1).

## 3.2 VHR dataset

The VHR dataset comprises high-resolution satellite orthoimages acquired at irregular intervals between 2000 and 2020 (Fig.
2b). The images in this dataset comprise a combination of data from three different sensors, namely aerial (0.5 m), Geoeye
(0.5 m), Pleiades (0.5 m). In this dataset, the panchromatic images have been orthorectified and resampled within the same
grid with a spatial resolution of 1x1 m. Given the variable spatial coverage of the VHR datasets, we have selected two sub-
areas (i.e. Tapado and Largo RG sub-regions, respectively; Fig. 2) where the largest amount of imagery is available,
attempting to maintain temporal coverage comparable to that of the L7/8 dataset.

With regard to the processing of the VHR dataset, it should be noted that the majority of the images were already
orthorectified and used directly from Robson et al., (2022), with the exception of two cases: (i) the photogrammetric flight in
2000's and ii) the 2014 Pleiades acquisition. Regarding the 2000's photogrammetric flight, we undertook a reprocessing of
the data by extending the area to the Largo rock glacier (4 km north from Tapado complex area) which was not initially
covered in Robson et al., (2022). The photogrammetric processing was based on the method set out by Cusicanqui et al.,
(2021) using Agisoft Metashape software v. 2.0.3 (Smith, 2011). Sixteen ground Control Points (GCPs) were used around
both small sub-areas. The 2019 Pleiades DEM from Robson et al., (2022) has been employed as a reference for the GCPs. A
coregistration stage using Nuth & Kääb (2011) has also been undertaken to correct small shifts of the 2000's aerial DEM.

With regard to the Pléiades 2014 acquisition, we applied the same methodology described in Cusicanqui et al., (2023) to
process the Pléiades 2014 stereo pair without GCP's and only Rational Polynomial Coefficients (RPC). The 2014 stereo
DEM was coregistered on the 2019 Pléiades DEM from Robson et al., (2022). Subsequently, the orthoimages were adjusted
in accordance with the aforementioned DEM co-registration values. For both datasets, images were acquired during the dry
season between November to April, spanning almost two decades (Table S1).





**Figure 2:** a) and b) Spatial extent and temporal distribution of L7/8 and VHR datasets, respectively; c) and d) Zoom over high resolution sub-regions used for validation. Orange polygons represent the 2013 rock glacier inventory from DGA, (2010) and yellow-dots represent the GNSS network on the Tapado complex (CEAZA, 2023). Image backgrounds correspond to © OpenTopoMap and Pléiades 2019 imagery © CNES/AIRBUS for c) and d).

### 3.3 Sentinel-1 interferograms

Due to the limited spatial extent of the VHR dataset, an inventory of gravitational movements, including rock glaciers and other periglacial processes, was compiled for this study using InSAR. This analysis was conducted through a visual inspection of several differential Sentinel-1 wrapped interferograms that covered the entire study area. S1 interferograms have been processed using the ForM@Ter LArge-scale multi-Temporal Sentinel-1 InterferoMetry processing chain – FLATSIM– service (Thollard et al., 2021), at different temporal baselines i.e. 12, 60 and 360 days. For this study, we employed 40 interferograms from the summer 2022 until summer 2023 in both, ascending and descending orbits (path 120



and 156, respectively) and created interferograms averaged in 2-looks (2 pixels in azimuth, 8 pixels in range) in radar geometry, 30 metres in terrain geometry. In brief, the FLATSIM service systematically produces interferograms from Sentinel-1 data and displacement time series, over large geographical areas. This service is based on the InSAR "New Small temporal and spatial BASelines" (NSBAS) processing chain as described in Doin et al., (2011) and Grandin (2015). FLATSIM products were corrected topographically using a SRTM-DEM and atmospherically corrected using ERA-5 atmospheric model mapped on the DEM. Full details can be found in Thollard et al., (2021) and ForM@TER platform. Raw InSAR wrapped interferograms were used for validation of rock glacier detection from L/78 imagery (cf. Section 4.1). It must be noted that this analysis was conducted only over a limited time-period (2023), so that gravitational movements not active during that period could have been missed.

### 3.4 GNSS data

The surface kinematics of the Tapado rock glacier have been measured by the Centro de Estudios Avanzados en Zonas Áridas (CEAZA). This network consists of a survey of 61 points since 2009 using a differential GNSS (dGNSS) system (DGA, 2010). According to CEAZA (2012, 2016) and Vivero et al., (2021), the base station coordinates were fixed using the Trimble CenterPoint RTX post processing service, and the differential post processing of the GNSS raw data between this base and the rover GNSS antenna was conducted using Trimble Business Center (TBC, V.4) surveying software. The reported average horizontal and vertical precisions (95%) were 0.02 and 0.04 m, respectively. In order to address some inconsistencies on point locations i.e. points systematically shifted few metres in north-east direction, 14 groups of GNSS points corresponding to the same block and specific dates (i.e. 2013-12-11, 2022-04-06, 2010-12-06) were removed from the original dataset. The remaining dataset comprises 47 groups of points and has been employed primarily for the validation of surface velocity maps derived from both L/78 and VHR dataset (cf. Section 5.3). Additionally, as no GCPs exist for Largo rock glacier, we manually tracked pseudo control points on representative features clearly identified on the VHR dataset to compare with L7/8 dataset (cf. Section 5.3).

### 4 Methods

The methodology employed in this study is based on the feature tracking offset image correlation strategy, which involves the analysis of a large number of images available for a site. Subsequently, inversion of time-series techniques were applied to the correlated images in order to derive consistent surface displacement fields over time. Then, a medium-resolution DEM was used to identify persistent moving areas within the study region. Eventually, we validate the final surface velocity fields by comparing them to recent DGNSS measurements and feature tracking of both L7/8 & VHR datasets on two small sub-regions in the upper regions of the La Laguna catchment (i.e. Tapado region).



### 4.1 inversion of displacement time-series

A time-series of horizontal displacement fields was obtained for each time-series of L7/8 and VHR orthorectified images, following a similar approach developed in Bontemps et al., (2018), previously applied on slow moving landslides (Lacroix et al., 2019). In brief, the approach adapted to our datasets can be described as follows:

a)   Feature tracking image correlation was performed in all possible pairwise combinations and their permutations (i.e. forward and backward). Two different approaches/software were used for each dataset. Firstly, we used Mic-Mac (Rupnik et al., 2017) through the Normalised Cross Correlation (NCC) algorithms to correlate images within the L7/8 dataset. This software was selected due to its suitability for images with low radiometric contrast or for small objects (Lacroix et al., 2020b). Secondly, the Ames Stereo Pipeline (ASP) (Beyer et al., 2018) was employed to
correlate image pairs within the VHR dataset. In ASP, the More Global Matching (MGM) implementation (Facciolo et al., 2015) was used to perform image correlation. The MGM algorithm reduces the amount of high-frequency artefacts in textureless regions and produces smooth surface displacement fields. Image mismatches associated with georeferencing errors are minimised due to the pre-alignment strategy before the feature tracking stage. Both softwares presents an adaptive windows matching strategy corresponding to 3x3 for MicMac and 7x7
for ASP for the small ones.

    b)   In both cases, all pixels with low correlation coefficient values (CC < 0.6) and displacement magnitude > 120 m, were masked. Furthermore, an additional glacier outline masking step was applied to the VHR dataset, to avoid noisy displacement values due to glacier retreat. The Randolph Glacier Inventory (RGI v.6) was used as the source of glacier outlines (RGI Consortium, 2017).

c)   Additionally, median surface displacement value was subtracted from the total displacement fields on both east-west (EW) and north-south (NS) displacement maps for all dates.

    d)   For the L7/8 dataset, an additional step was required to remove the striping effect in image correlation due to the sensor inter-band misalignments (Ayoub et al., 2008; Leprince et al., 2008).

    e)   A least-squares inversion of the redundant system per pixel is applied on both EW and NS surface displacements
components separately, to exploit the redundancy of pairs (Bontemps et al., 2018). This step has shown to reduce the uncertainties by about 30% on coarse resolution images of SPOT 1-4 satellites. A weight strategy can be added to the different pairs during the inversion, to take into account for instance to surface-cover changes over time. Due to the arid and natural cover of our area of study, this weight is not used here.

### 4.2 Automatic extraction of persistent moving areas (PMA)

The time series of ground surface displacements from L7/8 images are then used to automatically extract pixels persistently moving in a constant direction, as this is expected for motions driven by gravity (rock glaciers, landslides,...). Briefly, this methodology, developed by Stumpf et al., (2017), proposes to use the direction coherence of the displacement (called the



vector coherence) with time to detect active pixels. Additional filtering was applied by using the topography (i.e. TanDEM-X World DEM with 12 m resolution) and the slope orientation to detect pixels consistent with gravitational movements.

Namely we remove pixels whose mean velocity vectors are oriented less than 45° from the slope direction calculated over kernel size of 200 m to take into account the relatively large-scale undulations of the topography. These relatively high parameters have been chosen after a series of trials, and take into account the lower resolution of the images used compared to Stumpf et al., (2017), and the presence of snow in high mountains that can alter the quality of the displacement fields. Following this pixel-based approach, isolated pixels are removed and connected pixels are grouped to form moving areas.

### 4.3 PMA characterisation using InSAR and high resolution imagery

As mentioned in Section 3.3, InSAR wrapped interferograms were used mainly for validation and characterization of automatic PMA detection. Rather than create a new inventory of moving areas, we checked all polygons resulting from PMA methodology (cf. Section 4.2). As suggested in Barboux et al., (2014) and RGIK, (2023), we used a combination of all available interferograms with high resolution Google Earth imagery to classify slope movements. During this analysis, we consider the PMA valid when the polygons overlap a clear InSAR fringe pattern at any interval (e.g. 12, 24, 48 60 and 360 days) on the interferograms. The analysis resulted in a binary class, 'valid' and 'non-valid'. Secondly, a simple geomorphological class based on high resolution Google Earth imagery has been assigned to each polygon. The geomorphological class reflects the landform overlapping the PMA polygon e.g. landslide, rock glacier or other. When no clear interpretation about the movement and the geomorphology could be assessed on either InSAR or Google-Earth basemaps, the PMA was classified within the 'other' geomorphological class i.e. valid PMA located not so far from the ridges and at the valley bottom. Finally, we also assigned a velocity class for each PMA, based on RGIK, (2023) recommendations (cf. Section 6.2).



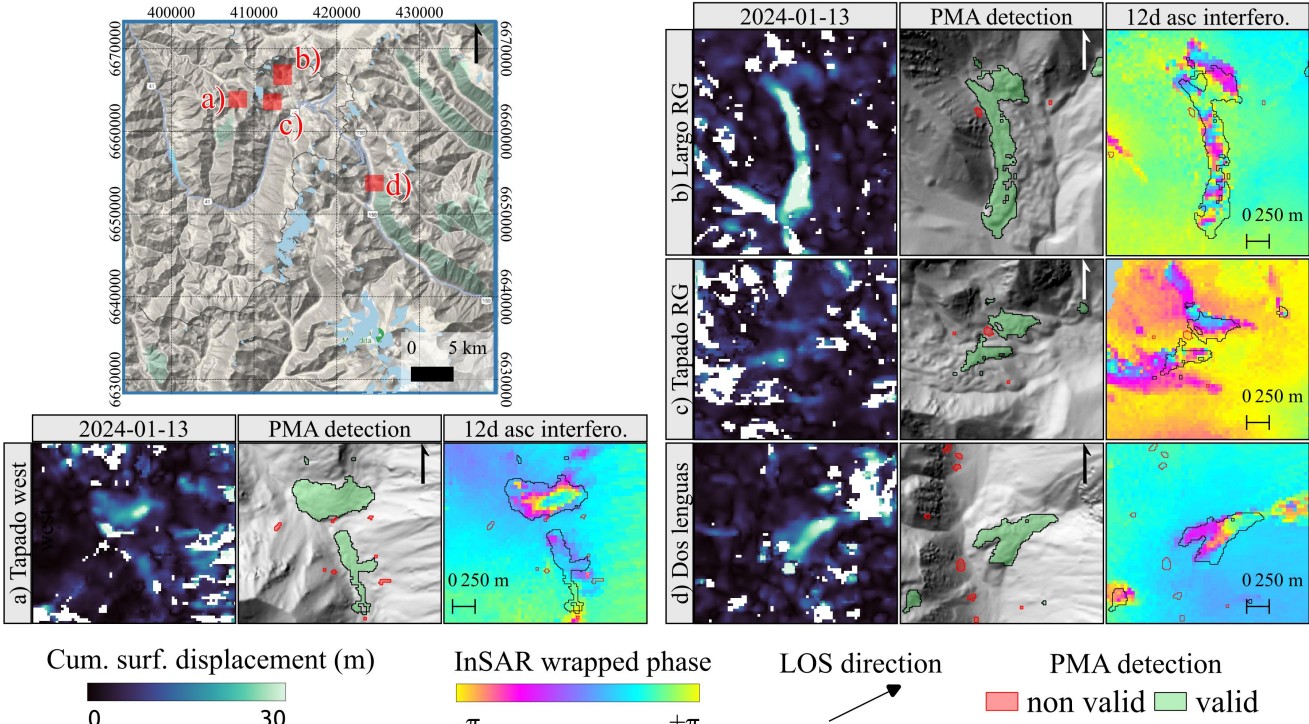

**Figure 3:** Example of raw outputs from inversion time-series, PMAs detection and PMAs validation using InSAR wrapped interferograms. Upper left map shows the location of small inner maps a) Tapado west,b) Largo RG, c) Tapago RG and d) Dos Lenguas. Image background corresponds to © GoogleTerrain. For all inner maps from left to right, show cumulative surface displacement map after inversion time-series last date available. In the middle, PMA's detection after directional and magnitude filtering and at right, 12 days ascending S-1 wrapped interferograms. Red and green polygons represent raw 'non valid' and 'valid' PMAs.

## 5 Results

### 5.1 Subsection (as Heading 2)

Within the area of interest covered by the L7/8 dataset the automatic PMA detection has produced 1710 polygons of moving objects. The raw PMAs area ranges from 225 to ~755,000 m² for the biggest one (Fig. 4). Those PMA's represent the group of pixels having a coherent movement in time and following the slope, thus likely gravitational movements. All PMA were verified employing the InSAR and optical cross-check validation detailed in Section 4.3. From this analysis, 29% of PMA were classified as 'valid' PMAs (nb = 501). Within the 'valid' PMAs, we classified 42% of the PMA as rock glaciers, 32% as landslides and 26% polygons classified as 'other'. Among the 42% of rock glaciers, we identified six rock glaciers directly connected to a debris-covered glacier. We decided to keep those within the 'rock glacier' class rather than create a





separate class because the PMA essentially covers the rock glacier component. Table 1 summarises all features and classes identified through the interpretation analysis.

**Table 1:** Summary of raw PMA geomorphological characterisation through cross-check verification using S1 InSAR and Google Earth optical imagery (cf. Section 4.3). Information about their statistical distribution could be found in Figure S1.

| TOTAL POLYGONS | | N | % |
|---|---|---|---|
| **Validity class** | **Geomorph. Class** | **1710** | **100** |
| **NON VALID** (not detected by InSAR on 2023) | **Sub total** | **1209** | **71** |
| | other | 747 | 62 |
| | valley bottom | 159 | 13 |
| | ridges | 155 | 13 |
| | landslide | 17 | 1 |
| | rock glacier | 15 | 1 |
| | glaciated | 116 | 10 |
| **VALID** (also detected by InSAR on 2023) | **Sub total** | **501** | **29** |
| | rock glacier | 211 | 42 |
| | landslide | 160 | 32 |
| | other | 130 | 26 |

On the other hand, 71% of PMAs (n = 1209) were classified as 'non valid' because no clear interpretation could be obtained from Google Earth optical imagery and interferograms. Within the 'non valid' class, we have noticed the presence of an

important number of small and isolated polygons (Fig. 4) located close to the mountain ridges and at the valley bottom (Fig. S3). As these tiny polygons cannot be correctly interpreted, we settled a first threshold based on the area of the polygons to remove those PMAs whose interpretation was ambiguous (Fig 4). All PMAs with less than 2250 m² (i.e. 10 pixels) have been removed from the analysis. The threshold of 10 pixels has been selected on the basis of PMA size and their corresponding InSAR fringe pattern (cf. Section 3.3), becoming difficult to interpret below this threshold. By using this

surface threshold, 43% (n = 735) of the entire PMA could be easily discarded. In addition, the 10% (n = 116) PMA were directly removed from those PMA corresponding to the deglaciated class.

The automatic surface threshold allows us to remove noisy values by only compromising 15% of valid PMA (Fig. 4), likely the most smaller (Fig. S3). After the surface threshold is applied, the remaining filtered dataset containing 901 PMA (47% of



the initial dataset), only 42% (nb = 382) of PMA are valid. Those PMA correspond mostly to large gravitational mass

movements among rock glaciers, landslides and other mass movements (Tab. 1). The mean surface of ~30,000 m² equivalent

to 134 pixels has been clearly identified on optical and InSAR images (Fig. S3). The remaining 68% (nb = 519) of non valid

PMA also represent a consistent group of pixels which potentially represents a gravitational movement, but these could not

be validated within the cross-check methodology using InSAR and interpretation of VHR imagery. Those have a mean area

size of 8,000 m² equivalent to 35 pixels. These polygons are often isolated and located close to the mountain ridges, or at the

valley floor. We could detect a few rock glaciers and landslides within the non-valid PMAs in the Google Earth imagery

(Tab. 1; Fig. S3), however it is very likely that they are inactive. From Figure 4 we can also state that the ratio between valid

and non-valid PMA's increases when PMA are bigger, suggesting that larger the object size, the higher the likelihood of

PMA detection using the L7/8 dataset. Further discussion regarding the possible causes of these polygons can be found in

Section 6.2. For the rest of the manuscript, we will only take into account the 382 valid polygons (i.e. after applying the

surface threshold).

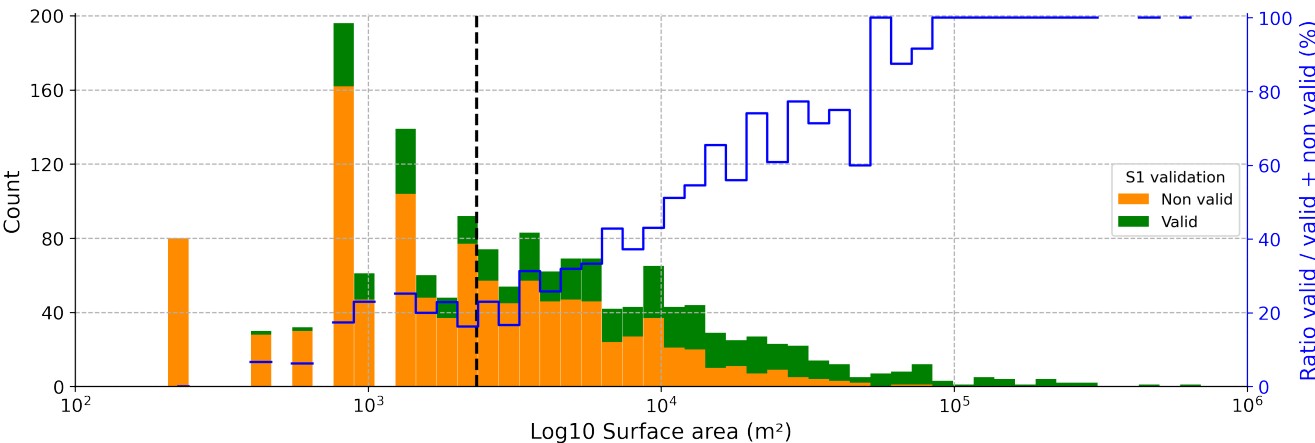

**Figure 4:** Distribution of raw 'valid' and 'non valid' PMA by surface area (bins = 50). Black vertical line represents the surface threshold i.e. 2250 m² (10 pixels) used as a filter to remove noisy values. All polygons at left of this threshold were removed. Blue line, represents the ratio between valid features over total features by bins. For access to our PMA polygons

for our own assessment, refer to the Data availability section.

**5.2 Regional distribution of surface kinematics**

Figure 5a provides an overview of the mean annual velocity over the 24 year period within the central Andes region. For

each PMA, we obtain a coherent downward surface velocity field overlapping a sector of a rock glacier (cf. Section 6.3 for

discussion). A first average velocity of 0.23 m yr⁻¹ over 24 years of displacements for all 382 PMAs was obtained. The

Normalised Mean Absolute Deviation (NMAD) computed over stable areas corresponds to ±0.07 m yr⁻¹ over 24 years. Refer



to Section 6.3 for a detailed discussion about the uncertainties. Stable areas were defined slopes lower than 35°, without taking account neither glacier outlines and all PMAs, also non-valid ones (Fig. S4). However, the representativeness of averaging surface velocity over the entire PMA surface can be questioned. As shown in Fig. 5 b to g, the pixels located in the borders often have values close to 0 m $yr^{-1}$, due to window sizes of feature-tracking algorithms. So, the boundary effect for

each PMA can bias the average velocity. To mitigate this bias, we propose to keep only the Top 50% fastest pixels within each PMA (hereafter referred to as 'Top 50% average velocity') to represent the average velocity for each PMA. By applying this criteria, the average velocity for all PMAs is $0.30 \pm 0.07$ m $yr^{-1}$ over the 24 years (Fig. 5a). Refer to Section 5.3 for a detailed discussion and a quantitative analysis about the influence of boundary effects.

Regarding the Top 50% average velocities independent for each geomorphological class –rock glaciers, landslides and

others– they correspond to 0.37 m $yr^{-1}$, 0.20 m $yr^{-1}$ and 0.18 m $yr^{-1}$, respectively. Among the classes, rock glaciers clearly show median average velocity faster (+23%) than the average velocity for the entire PMA dataset. Regarding the distribution of mean velocity, only 3 PMA have average velocities greater than 2 m $yr^{-1}$, corresponding to the Largo rock glacier (Fig. 2c; Fig. 5c), Olivares and Olivares west complex rock glaciers (Fig. 4f and g) and one landslide. Only 8 PMA have average velocities between $1 - 2$ m $yr^{-1}$, corresponding to 5 relatively large rock glaciers and 3 landslides. The rest of the PMA

dataset (n = 370) has average velocities below 1 m $yr^{-1}$.

In addition to the average velocity field, we were also able to obtain cumulative displacement time series (Fig. 5) of all PMAs (Fig. S9) over 24 years. The displacements time series are useful to depict temporal changes such as changes in velocity, e.g. accelerations or decelerations (Fig. 5e and f). For most of the rock glaciers (those with mean velocities > 1 m $yr^{-1}$), we can observe mostly a linear trend of surface displacements (Fig. 5b and d). Depicting annual velocity changes is

rather challenging because the average NMAD for all individual velocity pairs on stable areas corresponds to 1.18 m $yr^{-1}$. In some cases, particularly for the fastest and biggest PMAs, some accelerations (Fig. 5e) and deceleration (Fig. 5f) could be qualitatively assessed. This is evident because the fastest PMAs have had a total surface displacement between 40 to 70 m in 24 years, roughly equivalent to $2 - 3$ m $yr^{-1}$. In contrast to the medium sized PMA which have total surface displacements between $20 - 30$ m of displacement in 24 years, roughly equivalent to $1 - 1.5$ m $yr^{-1}$. Given the large uncertainties at annual

scale, the observed accelerations are not statistically significant for most of the PMAs.



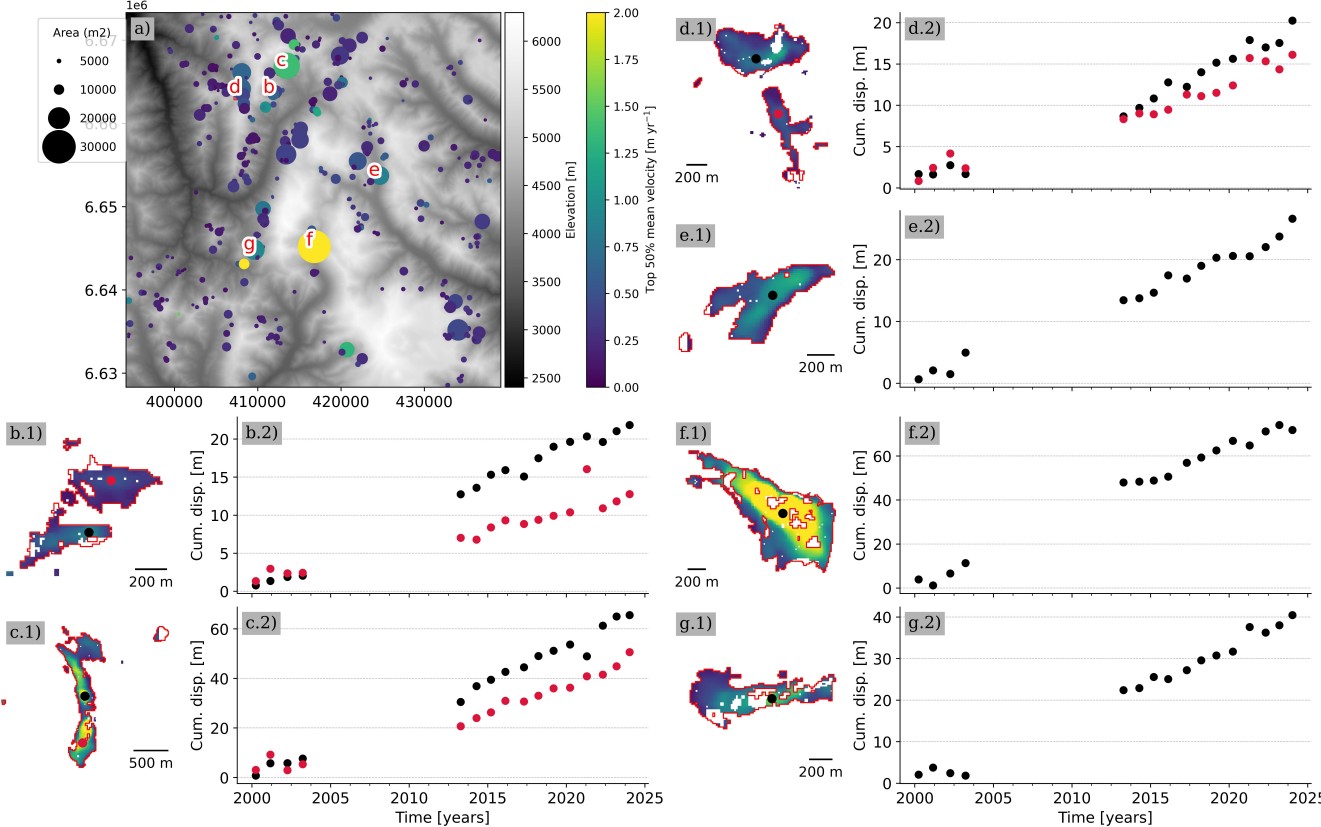

**Figure 5:** Surface kinematic characterisation for all PMAs in the central Andes region. a) Illustrates the spatial distribution of all valid PMAs (rock glaciers = 146; landslides = 115; others = 103) coloured by the 'Top 50% average velocity' surface velocity (viridis colorbar) within the PMA surface. The size of the circle is proportional to the rescaled PMA surface in m²/1,000 for better visualisation. The Red letters correspond to the study cases presented in the following subplots. The remaining subplots b) to g) illustrate mean annual velocity field over the 24 year (2000-2024) for a specific landform (subplots with a suffix of *.1) as well as the corresponding cumulative displacement time series in metres (subplots with a suffix of *.2), extracted on the black (and red) point within the landform. Subplots b) to g) correspond to the following landforms b) Tapado Complex and Las Tolas Rock Glacier; c) Largo Rock Glacier; d) Tapado west Rock Glacier; e) Dos Lenguas Rock Glacier; f) Olivares Complex, g) Olivares west Rock Glacier.

lthough annual velocity changes are challenging to identify, we also utilise 24 years of surface displacements to illustrate decadal velocity changes. In this context, surface velocities are computed over two larger periods, namely 2000–2014 ($V_1$) and 2013–2024 ($V_2$), for all PMAs. In this study, the relative changes in velocity are considered, with the first period serving as the reference (Eq. 1).





$$V_{change} = \frac{V_2 - V_1}{V_1} \, , \tag{1}$$

The NMAD values of decadal velocities over a stable area corresponds to 0.12 m yr$^{-1}$ for 2000–2014 period and 0.13 m yr$^{-1}$ for 2013–2024 period. The 'Top 50 average velocity' is 0.3 m yr$^{-1}$ at decadal scale corresponds to 43% of the overall average

velocity. Finally, assuming that NMAD for both periods are similar ($\sigma V$), the uncertainties related to the relative velocity change can be calculated using Equation 2. From Equation 2, it is possible to obtain a pixel-based uncertainty for the entire PMA surface. It can be observed that the greatest uncertainties are located on the borders of the PMA polygon, gradually decreasing as we approach the centre of the PMA.

$$\sigma V_{change} = \left( \frac{V_2 + V_1}{V_1^2} \right) * \sigma V \, , \tag{2}$$

As velocity change is dependent on both periods, it can be observed that uncertainties are larger when the magnitude of the velocity is small. Conversely, when the magnitude is large, the resulting uncertainties are relatively lower. To illustrate, consider a velocity change of 100% from period 2000–2014 to 2013–2024, respectively. An increase in velocity from 0.5 to 1.0 m yr$^{-1}$ is accompanied by an equivalent uncertainty reaching 0.78 m yr$^{-1}$, which represents 78% of uncertainty. Conversely, when velocity increases from 1.0 to 2.0 m yr$^{-1}$, uncertainties are 0.39 m yr$^{-1}$, representing 39% of uncertainty on

the relative velocity change. Based on the previous analysis, we can thus only rely on those PMA who have velocities greater than 1 m yr$^{-1}$. In this sense, only 3% of PMA are good candidates to obtain velocity changes statistically significant with lower uncertainties. Those PMAs correspond to 7 'rock glaciers' and 4 'landslides'. Computing average velocity changes in this filtered population independently for each class, rock glaciers have increased their velocity by +10% on average, the average velocity change is 1.03 to 1.06 m yr$^{-1}$ in two decades. In contrast, the velocity changes observed in 'landslides' up to

213%, passing from 1.08 to 2.03 m yr$^{-1}$ in average. Refer to Section 6.5 for further discussions.

**5.3 Velocity validation using GNSS and VHR datasets**

Once general characteristics of PMA have been described, we proceeded to compare surface velocity fields in more detail for the two selected sub-regions around the Tapado and Largo rock glaciers (Fig. 2a). A first comparison was made between ground truth points distributed along the main tongue of Tapado complex and Largo rock glacier and surface average

velocity fields. As shown in Figure 6, most of the ground truth points are located in the central flow line, with few points in the borders. This point to pixel comparison is shown in Figure 7 from where a good agreement between VHR and ground truth datasets can be seen. Some slight differences (i.e. underestimation of mean velocity) could be also observed notably on those points located on the borders of the Tapado complex as well as on some of the fastest points on the Largo rock glacier. Regarding the comparison between L7/8 and ground truth datasets, in the Tapado complex, both average velocities agree

relatively well (Fig. 7), with the exception of those points located at the borders. For the Largo rock glacier, the difference in





average velocity is more important because the average velocity field is more heterogeneous. The points located in the borders do not fit with the ground truth (cf. Section 6.3 for a detailed discussion).



**Figure 6:** Comparison of mean annual velocity over the 2000-2020 period for Tapado complex a) and b); and Largo rock glacier c) and d) for both L7/8 and VHR dataset, respectively. Red points show the location of ground truth, GNSS for Tapado complex (CEAZA, 2023) and pseudo-GCP for Largo rock glacier. White polygons correspond to their respective PMA identified from the L7/8 dataset (cf. Section 3.5).

Quantitatively, the average differences in velocity between VHR and ground truth points is about $0.01 \pm 0.05$ m yr$^{-1}$ (Tapado complex) and $0.38 \pm 0.3$ m yr$^{-1}$ (Largo rock glacier) while, the average difference between L7/8 and ground truth points is



0.18 ± 0.24 m yr⁻¹ (Tapado complex) and 1.35 ± 0.84 m yr⁻¹ (Largo rock glacier) (Figure 7). The good agreement on slow surface velocities on the Tapado complex could be explained by the homogeneous surface velocity field in both datasets (Fig. 6a). However, this is not the case on the Largo rock glacier where significant gradients in the velocity fields were found. This large difference could be likely explained by the heterogeneous surface velocity field from L7/8. Figure 6c

shows a single PMA that could be either divided in two, splitting Largo rock glacier in two different units, with likely independent dynamics. This is not the case for the VHR velocity field, showing rather a more homogeneous spatial distribution of velocities (Fig. 6d). In addition, some ground truth points were drawn close to the borders, from where L7/8 tends to underestimate the surface velocities.

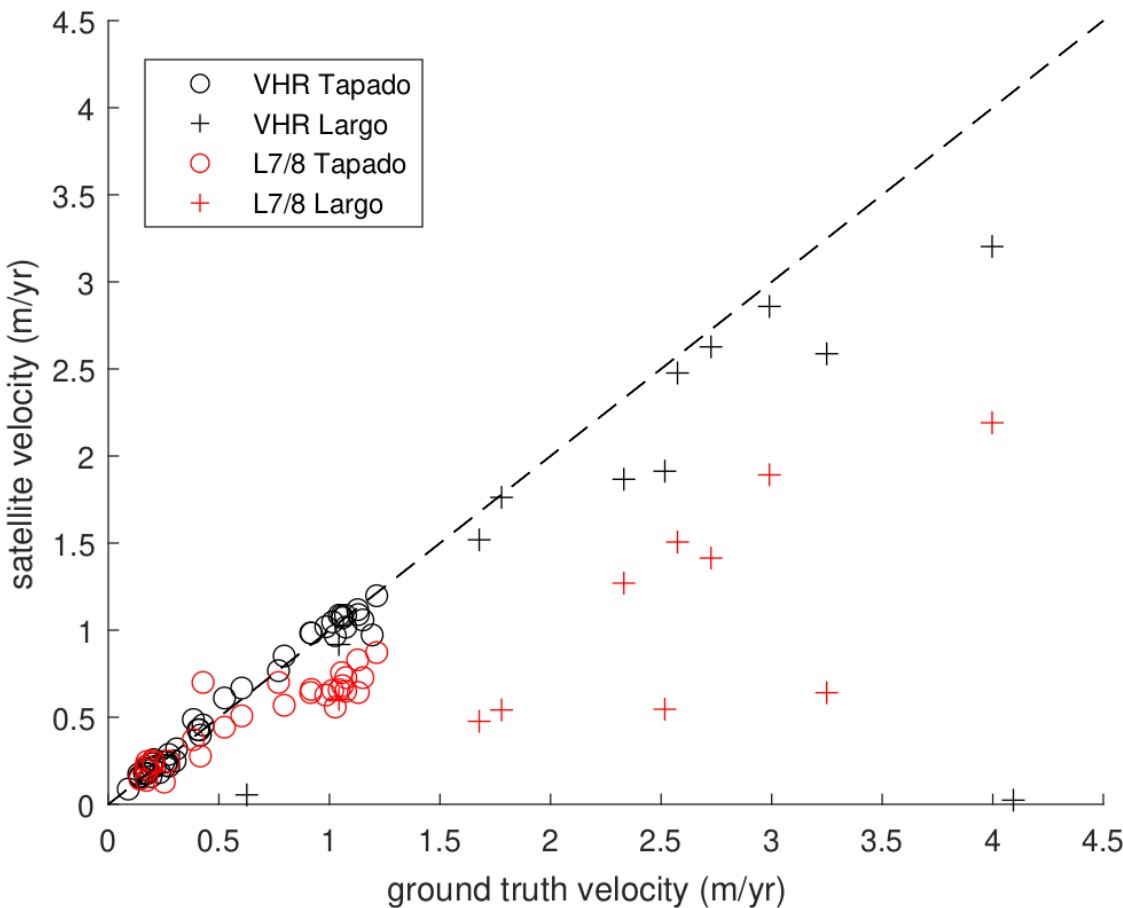

**Figure 7:** Point to pixel comparison between ground truth points and velocity fields from inversion time series for both

datasets in the Tapado complex and Largo rock glacier.

A detailed comparison with VHR optical imagery revealed a good agreement with ground truth data especially for the Tapado rock glacier (correlation coefficient of 0.99 and 0.45 for the Tapado and Largo rock glaciers respectively and linear



fit between VHR and ground truth of 0.99 and 0.44 over the Tapado and Largo). The correlation between L7/8 and ground
truth data is also very good, especially for the Tapado  (0.92 and 0.7 for the Tapado and Largo respectively) but, L7/8 tends
to always underestimate the velocities (linear fit of 0.69 and 0.45 for the Tapado and Largo rock glaciers).

In terms of velocity magnitude, the Tapado and Las Tolas rock glaciers do not present large differences. The L7/8 dataset
tends to slightly underestimate surface velocities up to 20% over those landforms. However, Figure 7 shows that when
looking at the highest velocities an important underestimation is observed. According to our estimations, this
underestimation can achieve between 10 – 20% on velocities up to 2 m yr$^{-1}$ and up to 30 – 40% on velocities above 2 m yr$^{-1}$.
Please refer to Section 6.1 for further discussion about the possible causes of the underestimations.

## 6 Discussion

Although rock glacier velocities are commonly estimated using high resolution optical data (e.g. Pellet et al., 2022) and SAR
remote sensing imagery (Strozzi et al., 2020, Villarroel et al., 2018), both datasets are relatively contemporary covering the
last 20 years (Toth & Jóźków, 2016). Those datasets are often limited in accessibility and can be prohibitively expensive for
larger areas. In this context, utilising archives of freely accessible L7/8 imagery emerges as a valuable source of information
for studying rock glacier dynamics over large spatial and temporal scales (Lacroix et al., 2020b), with a global reach. To the
best of our knowledge, this is the first time that Landsat imagery is being employed to monitor rock glacier displacement
time series and derive velocities. This is only possible through the combination of several robust methodologies, including
redundancy of information, inversion of time series and the persistent moving area detection. These allow the use of L7/8
data for rock glacier monitoring. However, several aspects must be discussed in order to account for the limitations, but also
perspectives of the use of Landsat imagery for RG kinematics analysis.

### 6.1 Intrinsic limitations on the remote sensing datasets

The initial technical point for consideration is the spatial resolution of the L7/8 dataset (i.e. 15 m in the panchromatic band),
which is pivotal when conducting feature tracking image correlation. Given that the pixel size is relatively coarse in relation
to the average surface velocity in the region i.e. ~1 m yr$^{-1}$ (Vivero et al., 2021; Halla et al., 2020), the method is expected to
only be appropriate for large and fast rock glaciers. This means that for regions containing large rock glaciers, such as the
Andes or High Mountain Asia (Sun et al., 2024), utilising medium-resolution imagery such as L7/8 offers new insights into
the temporal dynamics of rock glaciers. This rough hypothesis is based mostly on the number of pixels covering the
landform on the L7/8 imagery, which is certainly well suited for the Andes, but perhaps less so for other areas like the
European Alps, which contains generally smaller rock glaciers that may fall below the detection threshold. Furthermore, at
15 m of spatial resolution, distinguishing finer details is rather challenging. Figure 6 provides a clear illustration of the
impact of pixel size on the delineation of the rock glacier boundaries. The Tapado rock glacier, for instance, is not readily
distinguishable due to the coarse pixel size (Fig. 6a). This is particularly evident in the second tongue of the Tapado



complex, which is located directly south of the main tongue (see Fig. 2c). According to Vivero et al., 2021, the average velocity of the secondary tongue is between 0.25 – 0.5 m yr$^{-1}$. However, in our dataset, this is a noisy area with some gaps in the middle (Fig. 6a). This is also the case for Las Tolas rock glacier (which is contiguous to the Tapado rock glacier) which is also not well delimited (Fig. 7a). Nevertheless, despite the lack of clarity in the raw displacement fields, the automatic detection based on directional pixel filtering (Section 3.4) was able to successfully identify a coherent PMA over a

substantial area on the Las Tolas rock glacier, encompassing the majority of their tongue (Fig. 7a). This shows the potential of this filter for the detection of active RG, even at high altitudes where snow and shadows induce noise in the image correlation products.

It is also important to consider the roughness and texture of the rock glacier surface, which present less detail on L7/8 than the VHR dataset. The ridges and furrows morphology (Fig. 2c) can enhance/difficult image matching due to self-similar

features and could also play a role in ensuring a good feature tracking performance (Kääb & Heid, 2012). This is exemplified by the case of the main tongue of the Tapado complex (Fig. 6a), where the surface velocity field is comparable to the ground truth (0.01 ± 0.05 m yr$^{-1}$). The 24-year average surface velocities are in agreement with the results found by Vivero et al (2021). A comparison of the results presented in Vivero et al. (2021) with those obtained in the present study reveals a discrepancy of approximately 0.1–0.2 m yr$^{-1}$, which is likely attributable to the use of L7/8 images. Similar differences could

be observed on the Dos Lenguas rock glacier who presents an average velocity of 1.5–2 m yr$^{-1}$ according to Halla et al (2020) and Strozzi et al., (2020) while this study shows average velocities between 1.1–1.5 m yr$^{-1}$ (Fig. 5e). In contrast, the case of the Largo rock glacier is more complex to explain. Notwithstanding the presence of ridge and furrow morphology at Largo rock glacier, the rock glacier surface exhibits a relatively homogeneous texture (Fig. 2d). The discrepancy observed between the L7/8 and VHS results may be attributed to the influence of homogeneous surface roughness, despite the rock

glacier speeds being within the range of 3–4 m yr$^{-1}$. Figure 7b shows a clear disagreement on average surface velocities between the L7/8 and VHR dataset on the Largo rock glacier. This difference could be explained by the pixel size of the L7/8 dataset whose texture of the rock glacier surface is not well defined. In addition, the internal variability of surface velocity within the landform can also affect the output. As mentioned by Kääb & Heid (2012) and Leprince et al., (2008), the choice of correlation parameters are key when performing image correlation (Rosu et al., 2015). As an exemple, the small

matching windows on L7/8 imagery covers 3x3 pixels, corresponding to a surface of 2025 m² (45x45 m), in contrast to the smallest window used on VHR imagery, covering an effective surface of 49m² (7x7 m). In this sense, the values assigned to each pixel is kind of an average of a small window size chosen for correlation. Thus, as Largo rock glacier contains relatively large variability inside the landform, it is not surprising to find some discrepancies between the products. Finally, changes in solar illumination can also induce variations of shadows that can alter the quality of the image correlation

products (Dehecq et al., 2015). For this reason, all the L7/8 images were acquired mostly in March (with a few exceptions acquired in January) to reduce uncertainties related to the illumination.



## 6.2 Validation of PMA using InSAR and local rock glacier inventories

The average velocity fields obtained from L7/8 repeat satellite optical satellite data in this study are found to be in reasonable agreement with Sentinel-1 radar interferograms and their interpretation. Figure 8 presents a visual comparison between
InSAR wrapped interferograms and PMA characterisation. As Sentinel-1 radar wrapped interferograms provide only one motion LOS component, any discrimination of lateral and vertical movements cannot be depicted with a high level of accuracy (Barboux et al., 2014). In contrast, the surface kinematics obtained from optical imagery provide the two horizontal components.

With regard to the validity of our binary classification (i.e. 'valid' and 'non-valid'), it is evident that the PMAs are based on
quantitative measurement, yet it is clear that our InSAR activity classes and the velocity class assigned to each PMA involves some uncertainty/ambiguity. Firstly, some of the PMAs cover complex landforms without sharp boundaries. A general geomorphological class was assigned to those PMA without any specific discrimination, including those of debris-covered glacier and rock glacier transitions (Harrison et al., 2024). Secondly, those PMAs classified within 'other' geomorphological classes lack clear geomorphological features to be fully interpreted. The activity characterisation using
InSAR proved to be a valuable tool, particularly in the identification of so-called 'false PMAs'. These were observed to be concentrated in areas of low relief, often in proximity to riverbeds, and in close proximity to ridges (Fig. S2). Those errors could likely be explained due to smoothed DEM used as slope direction reference, as well as the role of shadows on the L7/8 images. Some other PMAs were discovered in close proximity to human settlements, including mining sites, where no evidence of InSAR displacement was observed.

Furthermore, it is evident that the radar and optical data available at the time of classification, such as their spatial resolution, will have an impact on the assigned velocity class. The InSAR velocity class for each PMA has been assigned in accordance with the methodology outlined in RGIK (2023). However, as the FLATSIM interferograms have a coarser pixel size (i.e. 30 m) than interferograms presented in Strotzi et al (2020) or Bertone et al (2021), the fringe pattern is sometimes barely discernible. Furthermore, the interferograms reflect the LOS movements within the specified time interval in 2023 (e.g. 12
days, 60 days, etc), whereas the PMA represents a cumulative movement over 24 years. Consequently, the reliability of the velocity class may be subject to significant uncertainty (Fig. 8).

The rock glaciers are slightly better detected than other features like landslides. This can be due to the higher motion variability over time of landslides. Indeed, rock glaciers are viscous flows (Haeberli et al., 2006) that face changes of velocities over long periods of time (Lehmann et al., 2021). On the contrary, landslides can be influenced by seasonal and
transient patterns (Lacroix et al., 2020b).





**Figure 8:** Comparison between Sentinel-1 wrapped interferograms at 12 and 60 days of interval and average surface velocity fields, in two small regions in the Central Andes.

A comparison was conducted between the two existing rock glacier inventories for the Chilean (DGA, 2022) and Argentinian (IANIGLA, 2018) Andes, with regard to those PMAs corresponding to rock glaciers only (nb = 153). In this limited area of the central Andes, the rock glacier inventories collectively enumerate 315 rock glaciers, 80 and 235, respectively, in each country. By utilising Ch-Arg RoGI as a reference, it can be observed that 68% of the PMAs (nb = 104) intersects the existing Ch-Arg RoGI at an average of 30% of their surface area (Fig. S12). Yet only 20% of the overlapping PMA (nb = 20) overlaps the Ch-Arg Rogi by more than 50% of their surface. The remaining 32% of the PMA (nb = 49) comprises unmapped rock glaciers, identified through L7/8 optical imagery and validated through inspecting interferograms





(RGIK, 2023). However, this comparison is contingent upon the veracity of the Ch-Arg RoGI, which also contains certain ambiguities. As an illustration, the Chilean rock glacier inventory has been updated on two occasions, in 2013 and 2022 (DGA, 2022). The most recent update reveals an overrepresentation of the rock glaciers, as the latest version of the inventory also encompasses some sectors of the headwall (Fig. S12). This is not the case with the Argentinian rock glacier inventory, which has restricted the rock glacier limits to more conservative outlines. Neither of the two inventories has been updated yet in accordance with the RGIK guidelines (RGIK, 2023). Finally, with regard to the contemporary activity of the PMA classified as rock glaciers based on InSAR velocity class, it can be stated that 69% of them (nb = 105) were detected using interferograms at 12 days (Fig. 8f) which is likely to correspond to a velocity class between 30 – 100 cm a$^{-1}$ (RGIK, 2023).

## 6.3 Average surface velocity of PMA?

In light of the fact that we have compiled average surface velocity maps for 382 PMAs (comprising 153 rock glaciers, 124 landslides and 105 non-classified objects) we pose the following question: what is the most appropriate methodology for computing average surface velocity fields? The most common approach to obtain representative surface velocity values for rock glaciers is to utilise the most active portion, which is often situated in close proximity to the central profile (RGIK, 2023). This avoids the potential for lateral variability within the landform (Fig. 6). In other studies, for example Kääb et al., (2021), the authors employed a small area on the most active sectors to express the representative velocity for the entire rock glacier. Nevertheless, the selection of this 'active' area remains somewhat subjective and may vary between users. In other respects, Blöthe et al., (2020) proposed the selection of pixels at the 95th percentile above the limit of Detection (LoD) to remove the lateral effects. The study by Blöthe et al., (2020) utilised detailed high resolution images captured by the RapidEye constellation with a pixel size of 6.5 m that allows to obtain a large number of pixels by landform.

In our study, computing statistics per landform using the same threshold could be complex, due to the large pixel size of L7/8 imagery. If we use the threshold of 50%, the bias resulting from lateral variability is automatically removed and an area in the middle of the PMA is conserved, corresponding to the fastest area (Fig 6). The area obtained from this methodology is independent for each PMA area and is solely based on the average velocity within the PMA. Figure 9 also presents a quantitative estimation of the influence of selecting 'Top 50%, 30% and 10% average velocity. The mean difference between the average velocity over the entire PMA and the 'Top 50%, 30% and 10% average velocity' corresponds to 20%, 31% and 44%, respectively. These differences underscore the significance of selecting the most appropriate thresholds. Here, we consider the 'Top 50% average velocity' as it represents an optimal compromise between the remaining amount of pixels and the average velocity field computed without lateral effects, with the potential to reduce the ambiguity introduced by the operator. Nevertheless, further studies should be conducted to more rigorously evaluate this metric in diverse contexts and with disparate datasets.



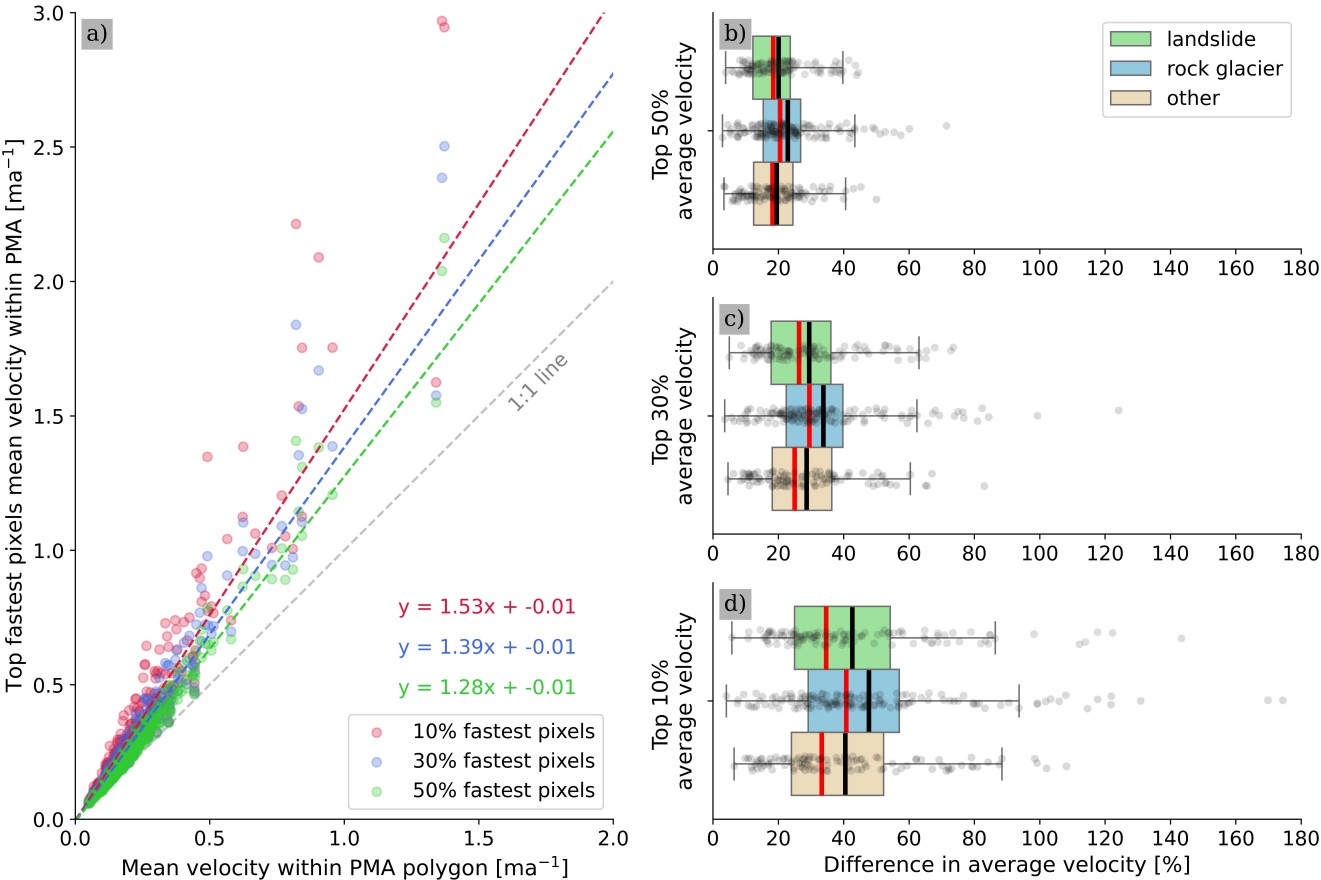

**Figure 9:** a) Comparison between mean average velocity computed using the entire PMA surface and 'Top 50%, 30% and 10% average velocity' within PMA. Subplots b), c), and d) show the difference of average velocity 'Top 50%, 30% and 10% average velocity' in respect to the velocity computed over the entire PMA surface.

## 560 6.4 Surface velocity and uncertainities

The methodology employed in this study using Landsat images permitted the acquisition of relatively low surface velocity uncertainties. In a first order, the pixel time series inversion (cf. Sections 3.2) demonstrates the potential to reduce uncertainties by approximately 30% on coarse resolution images, as reported by Bontemps et al (2018) and Lacroix et al., (2020a). Furthermore, the PMA methodology (cf. Sections 3.2) enable the removal of anomalous pixels, defined as pixels exhibiting incoherent movements in time, based on a directional threshold, while retaining only those pixels who present coherent movement (Stumpf et al., 2017). In conclusion, the uncertainties have been reduced solely to the mobile zones. By focusing on the uncertainties in those PMA where we have ground truth data, namely Tapado and Largo rock glacier, our methodological approach tends to result in a sub-estimation of surface velocities between 10–20% on average. However, as illustrated in Figure 7, the degree of underestimation is contingent upon the rock glacier types. Even when ground truth





datasets align well on Tapado surface velocities (between 1–2 m yr⁻¹), the Largo rock glacier velocities (2–4 m yr⁻¹) are underestimated by 30–40%. As previously mentioned (Section 6.1), we attribute these differences to the difference of textures of the two rock glaciers. However, there is a general underestimation of surface velocities using L7/8 imagery. This could be caused by the large pixel size (i.e. 15 m) that leads to few pixels on the landforms per matching window during correlation stage and to lower contrast. Furthermore, the lack of data between 2003 and 2013 could also lead to some bias on time series of displacements.

We managed to validate the velocities over two of the 382 PMAs with in situ measurements or VHR data. We can wonder about the validity of this validation for the remaining rock glaciers. Our calculations indicate that the NMAD over stable areas corresponds to 60% of the 45x45 km² (Fig. S4), which equates to 0.07 m yr⁻¹ ± 0.16 as standard deviation. A similar range of uncertainties was identified by Kääb et al., (2021) in the Tien Shan region, employing the same approach using high resolution historical images with poor scan quality and VHR imagery. In contrast, Blöthe et al., (2021) employed a distinct methodology based on the Limits of Detection (LoD), with uncertainties ranging from 0.28 to 0.5 m yr⁻¹. This was achieved through the utilisation of high-resolution optical images. The low uncertainty value observed in the L7/8 imagery can be attributed to the inversion time series pipeline (Bontemps et al., 2018), which enables the removal of anomalous pixel values within the time series based on the redundancy of information, thereby reducing the impact of errors on those pixels that represent anomalous behaviours. Furthermore, this value represents the average uncertainty over a period of 24 years.

### 6.5 Velocity variations?

The uncertainties on the velocity at annual scales are considerably greater than those estimated over the entire period. Our estimations indicate that the average NMAD over stable areas (Fig. S3) for all annual surface displacements is 1.67 m ± 0.33 (as standard deviation) and 1.18 m yr⁻¹ ± 0.44 (as standard deviation). These values are consistent with those reported by several studies (Bontemps et al., 2018; Lacroix et al., 2020a; Leprince et al., 2008; Scherler et al., 2008), who also employed L7/8 images. In accordance with the methodology proposed by Blöthe et al. (2021), the application of the mean velocity NMAD as a LoD filter to all PMAs within the region resulted in the retention of only 2% of PMAs (n = 8) above the specified threshold. These PMAs correspond to relatively large and fast rock glaciers, as illustrated in Figure 5c, f, and g. This analysis demonstrates that the use of L7/8 imagery does not yet permit the differentiation of annual velocity fluctuations in rock glaciers, but rather a kinematic characterization over 20 years. Further studies are required to enhance our understanding of the annual velocity changes observed employing L7/8 images.

Regarding decadal velocity changes between the two periods 2000-2014 and 2013-2024 (Eq. 1) questions arise as to whether the observed changes in velocity are sufficiently representative, given the relatively large uncertainties in velocities (Eq. 2). The answers to this question can be divided into three main aspects:

- One potential source of discrepancy in the observed velocities is the differences in the length of the 2 observation periods. The first period, spanning from 2000 to 2014, comprises only six years of observations due to a gap between 2003 and 2013. In contrast, the second period, spanning from 2013 to 2024, encompasses 12 years of



continuous observations. This may influence the average velocity for each period and thus condition the related uncertainties (Fig. 5). The use of other medium resolution data like ASTER could be used to fill this gap between 2003 and 2013, despite the low radiometric resolution of the data.

- The second element is the size of each PMA. L7/8 imagery tends to perform better on bigger landforms due to the higher amount of moving pixels. Nevertheless, high uncertainties on velocity change are present on borders of PMA due to the small velocity magnitude, on both periods (cf. Section 5.2). The border effect is less pronounced on bigger PMAs than the smaller ones. However, the bigger PMA obtained in this study are landforms related to complex processes (glacier-permafrost interactions) which may have been influenced by internal landform variability. This is the case of Largo rock glacier, which experiences a velocity change of +54% in the L7/8 data and +29% in VHR. In contrast, in the case of Olivares complex (debris-covered glacier connected), the mean velocity change corresponds to -9% in the L7/8 data. Similar findings were observed in the Tien Shan region (Kääb et al., 2021) and in the French Alps (Cusicanqui et al., 2023) showing that these complex interaction may have an influence on surface velocities of contiguous landforms, highlighting complex processes still poorly understood. Finally, the size of PMA is not a limiting factor when high velocity changes are observed.

- The few observations of velocity changes in the Andes show few variations over the last 20 years. Vivero et al., (2021) in the Tapado rock glacier shows 7% of acceleration in the same study period. Our VHR data presented in this study show limited variation in velocity between 2000-2010 and 2010-2020 with a slight slow-down of the Tapado rock glacier by -11% and a speed-up of the Largo rock glacier by +14%. The longer dataset presented in Vivero et al. (2021) shows larger variations over 40 years with an acceleration of 0.2 m yr$^{-1}$ of the Tapado rock glacier between 1980–2000 and 2000–2020 period, representing 25% of velocity increase in average. Such a level of acceleration might not be detected by medium resolution imagery. It would therefore be beneficial for future studies to incorporate additional older datasets, to extend the time series with medium resolution satellite images, e.g. SPOT 1-4 up to the mid 1980's or Corona images from the 1960s (Dehecq et al., 2020).

### 6.6 Controlling factors of PMA spatial distribution

With regard to the topographic conditions on which the PMAs are located, Figure 5a illustrates the heterogeneous spatial distribution of PMAs within the study area. In order to illustrate the relationships with topographical characteristics and velocity, we compare Top 50% average velocity with mean slope, aspect, elevation and surface area, all of which were extracted using the TanDEM-X 12.5 m DEM (Fig. 11, Fig. S7). A number of noteworthy observations emerge from this comparison.

- The rock glaciers are uniformly distributed on slopes between 10–35°, whereas 'landslides' and 'other' features have modal distributions that peaks at 30° and 35° slope, respectively, and not situated on slopes less than 25°. This is in agreement with the regional slope of the study area suggesting a control of slope for landslides and others (Fig. 11a).



- Distributions of 'rock glacier', 'landslide' and 'other' features slope orientations show contrasted distributions. On one hand, rock glaciers are predominantly on West to South and East slope faces. Our findings are in agreement with regional Permafrost models (Gruber, 2012, Obu, 2021; Azocar et al., 2017). On the contrary, landslides and others are predominant on North-west to East slope faces (Fig. 11d). These results suggest possible correlations between the response of gravitational movements in high mountain areas (e.g. Haeberli et al., 2017; Patton et al., 2019) to permafrost degradation i.e. freeze, thaw of permafrost. Similar results were found by Blothë (2021) in the Cordon del Plata highlighting the role of slope orientation as a controlling factor.

- With regard to the altitude, most of the rock glaciers with velocities between 1–2 m yr$^{-1}$ are concentrated at higher altitudes (4,500–5,000 m a.s.l.). Our findings do not suggest a clear correlation with altitude for rock glaciers. This is probably not the case for the PMAs identified as 'landslide' and 'others', which also occur at lower altitudes of 3500 m a.s.l. (Fig. 11b). Most of them are located on terrains where permafrost permafrost can be described as heterogeneous or discontinuous (Gruber 2012; Azócar et al., 2017).

- The 'Top 50% average velocity' revealed that rock glaciers with a larger surface area tend to have higher surface velocities, in contrast to the 'landslide' and 'others' categories, which do not exhibit the same behaviour. This could be likely the consequence of the local geomorphological context (i.e. Largo rock glacier) and the complex interaction with glacier and debris-covered glaciers (i.e. Tapado and Olivares complexes). Largo rock glacier (Fig. 5b) experiences a velocity change of +54% in the L7/8 data. It is plausible that the acceleration is a consequence of the extensive accumulation area at the summit of the Largo rock glacier (Fig. 2c). According to our interpretation, it supplies the rock glacier with a considerable quantity of material i.e. snow, ice, debris avalanches (Janke and Frauenfelder, 2008). On the other hand, the Olivares rock complex (Fig. 5f) provides an illustrative example of a rock glacier undergoing deceleration. Given the context of likely debris-covered glaciers connected to the rock glacier, this slowdown process could be the consequence of ice-mass loss. Similar observations have been made in the Tien Shan region (Kääb et al., 2021) and more recently in the European Alps (Manchado et al., 2024). Further studies should be carried out on the understanding of the mechanics of these complex landforms.

Finally, although this dataset is a valuable approach to depicting relationships between surface kinematics and topographic parameters at the regional scale, nevertheless, the current results should be interpreted with caution, as the current dataset presents morphological statistics computed solely within the PMA surface which does not necessarily represent the whole feature (Fig. S12) and particularly the upper part that could be covered by snow. Further studies should be conducted to include kinematic attributes within more accurately defined official regional geodetic information.





**Figure 10:** Comparison of the PMA distribution for 'rock glacier' (blue values), 'landslide' (green values) and 'other' (orange values) geomorphological class vs topographical context. a) PMA mean slope; b) distribution between Top 50% mean velocity PMA and PMA mean elevation; c) distribution between Top 50% mean velocity and PMA surface; d) PMA slope orientation. For a) and b), the grey background represents the general slope and elevation context of the study area, respectively.

# 6 Conclusions

This study presented a robust method to detect, quantify, and analyse the surface kinematics of rock glaciers and other gravitational slope movements using time series of Landsat 7/8 imagery. By integrating feature tracking from 24 years of imagery with time series inversion and automatic detection of persistent moving areas (PMA), the study successfully monitored 382 movements over a 45x45 km² area in the semiarid Andes. Validation using satellite radar interferometry confirmed the classification and velocity attributes of the PMAs, with 42% also detected by Sentinel-1 interferograms at 12



days temporal baselines. In addition, faster landforms (2–4 m yr⁻¹) were also detected, mostly corresponding to complex ice-debris landforms. The 24-year mean velocity was $0.3 \pm 0.07$ m/year, with rock glaciers moving 23% faster than the median of all types of geomorphological objects. Despite underestimations due to pixel size and temporal gaps of images, decadal

velocity changes were observable under certain conditions, notably when average velocities are greater than 1 m yr⁻¹. Below this velocity threshold, changes in velocity using L7/8 data are not statistically significant and could not be safely assessed. The study aligned well with existing research and highlighted the potential of integrating radar and optical remote sensing for enhanced interpretation, detection and monitoring of slow and fast gravitational movements. The findings support rock glacier mapping and understanding of their kinematics, especially in light of permafrost warming and its effects on

periglacial landforms. Finally, this study sets up a methodological benchmark demonstrating the feasibility of medium-resolution L7/8 images in quantifying the kinematics of rock glaciers and ice-debris complex dynamics at a regional scale. This study responds to the need of the scientific community to contribute to the assessment of the current state of periglacial landforms of surface deformation on different spatial scales, using worldwide available open-source optical imagery.

*Code availability.* Feature tracking image correlation softwares used for this study are open-source softwares. Ames Stereo Pipeline (ASP) is available from https://stereopipeline.readthedocs.io/en/latest/introduction.html (Bayer et al., 2018) and MicMac is available from https://micmac.ensg.eu/index.php/Accueil (Rupnick et al., 2017). Sentinel-1 interferograms were computed using ForM@Ter LArge-scale multi-Temporal Sentinel-1 InterferoMetry processing chain (FLATSIM) based on NSBAS pipeline. Both are available through GDM-SAR service at https://www.poleterresolide.fr/le-service-gdm-sar-in/.


*Data availability.* Landsat 7/8 archive freely available at http://earthexplorer.usgs.gov/. Sentinel-1 data used in our study are freely available from the ESA/EC Copernicus Sentinels Scientific Data Hub at https://scihub.copernicus.eu (Copernicus Open Access Hub, 2021). FLATSIM Sentinel-1 interferograms can be accessible upon request via Form@Ter pole (https://www.poleterresolide.fr/). TanDEM-X data are available from DLR through proposal application procedures. Data

from Digital-Globe satellites (GeoEye, Ikonos, WorldView, Quickbird) and Pléiades are commercial, but programmes to facilitate academic access exist. Pleiades dataset can be accessed upon request to Ben Robson (Benjamin.Robson@uib.no). The data described in this manuscript are available at XXXX (Cusicanqui et al., 2024) or upon request from the corresponding author (diego.cusicanqui@univ-grenoble-alpes.fr).

*Supplement.* The supplement related to this article is available online at: https://zenodo.org/uploads/13119042.

*Author contributions.* DC, PL and XB designed the study. DC performed image correlation of VHR data provided by BR and XB. PL performed image correlation of L7/8 data and implemented persistent moving area (PMA) detection. DC and PL filter GNSS dataset provided by SM and compute GNSS surface velocity time-series. DC wrote the paper with the

supervision and contributions of PL. PL, XB, BR, AK and SM contributed to the discussion and edited the paper.



*Competing interests.* The authors declare that they have no conflict of interest.

*Acknowledgment.* Thanks are due to reviewers for their careful review and comments. We are grateful to the providers of free data for this study: European Space Agency (ESA)/European Commission (EC) Copernicus for Sentinel-1 data, the
FLATSIM Form@TER team for their efforts processing Sentinel interferograms. Also, the German Aerospace Center (DLR) for TanDEM-X data. TanDEM-X data were provided by DLR. We are grateful to CNES/Airbus DS for the provision of the SPOT and Pléiades satellite to the restrained dataset project 41743. We would like to thank the U.S. Geological Survey for making the Landsat 7/8 archive freely available. Thanks to GLIMS database http://glims.org/RGI/ for glacier outlines (v.6). All (or most of) the computations presented in this paper were performed using the GRICAD infrastructure
(https://gricad.univ-grenoble-alpes.fr), which is supported by Grenoble research communities. Thanks to the glaciology group at CEAZA for collecting and providing the GNSS datasets, and the CHESS-funded (https://chess.w.uib.no/) "Summer school on cryospheric monitoring and water resources" for the 2022 acquisition set.

*Financial support.* This work has been supported by the postdoctoral program from the National Centre for Space Studies
(CNES) and partially by the National Center for Scientific Research (CNRS), the program Plan d'Action pour la Prévention des Risques d'Origine Glaciaire et périglaciaire (PAPROG). This work has also been partially supported by a grant from Labex OSUG (Investissements d'avenir – ANR10 LABX56) PermANDES project. Diego Cusicanqui (CNES | ISTerre), Pascal Lacroix (IRD | ISTerre), Xavier Bodin (EDYTEM | CNRS) are part of Labex OSUG (ANR10 LABX56).

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
