# Peer review of "Detection and reconstruction of rock glaciers kinematics over 24 years (2000-2024) from Landsat imagery"

_EGUsphere, 2024_

## Author Comment (AC1)

**Response to comments by first reviewer**

**Dr. Jan Henrik Blöthe**

We sincerely thank Dr. Jan Henrik Blöthe for the thoughtful comments and constructive suggestions, which have greatly contributed to improving the clarity and overall quality of our manuscript. Below, we provide a point-by-point response to each comment.

**General comments**

**GC-1.** In the introduction and the formulation of the aim of the study, the authors focus entirely on the rock glaciers and their surface kinematics. The justification of the study is well written and the work is a valuable contribution to the high-Andean permafrost research. That being said, the inclusion of landslides and other "gravitational slope movements" into the analysis introduces confusion and shifts the attention of the reader away from the subject of the manuscript. In my view, focusing the content and interpretation to the rock glaciers and summarizing different movements detected by the approach applied here as "other", or "non-rock glacier" would not only increase the clarity of the work, but also make the work more in line with the aim formulated at the beginning as well as the community addressed here. The alternative would be to include a larger paragraph on landslide processes in the region in the introduction, addressing different types of landslides found and discussing their mechanisms of movement.

**Response >** We agree with the reviewer when he asks for the inclusion of landslides and other "gravitational slope movements" in the introduction section. However, as the entire paper is focused on rock glacier surface displacements and velocities, we believe that introducing landslides and other landforms will introduce more confusion. In fact, landslides and other landforms have not been really exploited and discussed because it was not the focus of the paper. In this sense, we added a paragraphs in the discussion section to justify our choice. Now you can read:

*[...] Although the objective of our study is the monitoring of rock glaciers on a regional scale, other PMAs corresponding to mass movements such as landslides could also be identified. Our results suggest possible correlations between gravitational movements in high mountain areas (e.g. Haeberli et al., 2017; Patton et al., 2019) and permafrost degradation (i.e. freeze, thaw of permafrost) in recent deglaciated areas (Pánek et al., 2022). This study allows us to complete those existing mass movement inventories in the region (e.g. Iribarren Anacona et al., 2015), highlighting areas for further research. While [...]*

**GC-2.** I am very glad to see that the authors go beyond the idea of calculating mean velocities for rock glaciers. Despite the wide use of mean values for entire landforms, the very heterogeneous velocity distribution found in rock glaciers makes this very difficult. In my opinion, the authors propose a suitable metric in their analysis. However, in order to make their approach comprehensible and reproducible, a consistency of terminology is very important:

○      In L335-336, the authors propose to use the "Top 50% fastest pixels within each PMA" as this metric ("top 50%" and "fastest" are redundant here). I would recommend to either elaborate this more clearly in the text, or, preferably, include a small figure of one of the rock glaciers, illustrating what the idea behind this approach is. If I am not mistaken, the authors suggest calculating the mean of all velocity values between median and maximum, i.e. the upper 50% of the data distribution. If I am not mistaken, this metric is a "spatial mean", as opposed to the "average velocities" computed over different temporal intervals. Especially for the text in the sections 5.3 and 6.3 it is imperative to have an unambiguous terminology throughout.

**Response >** This is in fact one problem we faced for computing average velocities per landform. The idea behind Top 50% average velocity is to remove the effect of the borders and keep the fastest area based on the 24-year surface velocity map. The aim of this methodology is to reduce the ambiguity behind manual selection of the most active areas. That being said, we checked the consistency of the term 'Top% average velocity' throughout the manuscript specially when 'average' and 'mean' velocities are computed.

Regarding the suggestion of adding a new figure, we indeed added a new figure on the supplementary material (Figure S15). This figure is attached below. The reason behind it, is because the second reviewer asks for a specific section showing more in detail the computed uncertainties. So, we prefer to add a new table as well as a new figure (now Figure 8) showing the temporal evolution of the uncertainties.

[Figure]

Figure S15. Representativity of spatial average statistics for four landforms: (a) Olivares debris ice complex, (b) Tapado complex, (c) Largo rock glacier, and (d) Dos Lenguas rock glacier. Subplots labeled with a suffix *.1 illustrate the pixel distribution for each metric used in this study: All (blue), Top 50% (orange), Top 30% (green), and Top 10% (red) average velocity. Subplots with a suffix *.2 display boxplots representing surface velocity for each landform, showing the variability and representativity of the computed metrics. Red and black lines in the boxplot represent the median and mean values respectively.

**GC-3.** To allow the interested reader to follow the processing steps, and ultimately to allow the community to make use of the workflow, the processing chain outlined mainly in sections 4.1 and 4.2 should be explained in a bit more detail (also see specific comments on that sections below).

**Response >** We agree with the reviewer's observation. However, as the methodologies for time series inversion (Bontemps et al., 2018) and automatic PMA extraction (Stumpf et al., 2017) are already thoroughly detailed in dedicated papers, we have opted to retain only the essential aspects relevant to this study. Since the focus of this paper lies primarily on the methodological advancements applied to rock glaciers, we prioritize discussing key elements such as the comparison with very high-resolution datasets and the associated uncertainties, which are crucial for the interpretation and validation of our results.

**Specific comments**

**SC-1.** L14-15: This statement is a bit vague, as neither "long-term" nor "high-resolution" are clearly defined. Can you be more specific here?

**Response >** We modified the sentence to be more specific. Now you can read:

*[...] However, representing **decadal regional** spatio-temporal **velocity** patterns remains challenging due to **the limited number of** high-resolution **(< 5m)** remote sensing data. [...]*

**SC-2.** L18-19: From a geomorphic perspective, gravitational slope movement would not entail rock glaciers that are bound to the presence of permafrost.

**Response >** From a geomorphic perspective, we agree that gravitational slope movements encompass a wide range of processes. However, rock glaciers are distinct geomorphic features specifically associated with permafrost and frozen ground dynamics. Their morphology, internal structure, and movement patterns are directly influenced by the presence of ice within the subsurface, which differentiates them from other gravitational processes not tied to permafrost. Now you can read:

*[...] This methodology enables the detection and quantification of surface kinematics of 382 gravitational slope **mass** movements (153 corresponding to rock glaciers) [...]*

**SC-3.** L74: It could be stated a bit more clearly here, what "the current warming context" refers to. Also for the specific study area, rising air temperatures have been documented, as the authors describe later.

**Response >** We modify the sentence to be more clear. Now you can read:

*[...] Given the **observed** current warming context **of mountain permafrost (Noetzli et al., 2019)** [...]*

**SC-4.** L81-82: And for short observation periods, or high temporal resolution of SAR imagery.
**SC-5.** L83-84: As the Landsat 7/8 data used in the present study spans the same time interval, I don't think this statement adds to the justification here. As you are making this argument later on, why not focus this statement on freely available SAR imagery?

**Response >** Here we answer both previous comments. We modify the sentence to be more clear. Now you can read:

*[...] However, even if this technique is well suited for rock glacier mapping (Barboux et al., 2014) satellite radar interferometry is **most suitable** for relatively slow rock glacier speeds of **approximately** 1–1.5 m a$^{-1}$ **observed over short periods (6, 12 days)**. Beyond this threshold, InSAR signals become geometrically decorrelated and thus uninterpretable (Villarroel et al., 2018). In addition, **freely** SAR data with high temporal **resolution has** only **been** available since the early 21st-century (Strozzi et al., 2020). [...]*

**SC-6.** L95: As the authors are citing our work (Blöthe et al. 2021) later anyway, it deserves to be mentioned here as well, given its spatial focus.

**Response >** We added the citation corresponding to Blöthe et al., 2021 as well as more recent studies from Kääb and Røste, (2024). Now you can read:

*[...] some isolated regions in the Andes i.e. Tapado rock glacier (Vivero et al., 2021; **Blöthe et al., 2021**), in northern Tien Shan (Kääb et al., 2021) **and more recently in the United States (Kääb and Røste, 2024)**. [...]*

**SC-7.** L102-104: If it is really the aim of the study to investigate rock glacier velocities "for the late 20$^{th}$ century", imagery from 2000 to 2024 maybe wasn't the best choice :-)

**Response >** We clarified the text. Now you can read:

*[...] rock glacier displacements and velocity for the **early 21th** century in a region [...]*

**SC-8.** L122: Are these weather stations located in the study area? If so, please include their locations in Figure 1.

**Response >** We included the Automatic Weather Station (AWS) used in this study in Figure 1. Now the figure look as follow:

[Figure]

**SC-9.** L139-141: I am being a bit picky here, but our study investigates velocities between 2010 and 2017/18.

**Response >** Thanks for pointing that out. We corrected the text accordingly.

**SC-10.** Figure 1 / L148-152: As the map shows the Gruber 2012 model of permafrost distribution, I suggest the authors also use the terminology of Gruber in the legend presented here or explain how this was adopted.

**Response >** We changed the legend and adopted the original from Gruber, 2012. Figure 1 has been updated accordingly. Please refer to the response of the specific comment SC-8 to look at the updated figure.

**SC-11.** L159-162: It is somewhat confusing at this point… maybe first mention that for all analysis you were relying on the panchromatic band, before describing the ETM+ data gap.

**Response >** We modified the paragraph to be more clear. Now you can read:

*[...] **We used only the L7/8 panchromatic band (B8) with the highest spatial resolution (15 m). However, due to the** Scan Line Corrector **failure** on the **Landsat-7** satellite **between 2004 and 2013** (Markham et al., 2004), **we excluded Landsat-7 scenes within this period to avoid data gaps.** [...]*

**SC-12.** L191-192: The "landslide" and "other" classes as not necessarily periglacial processes.

**Response >** We modified the paragraph to be more clear. Now you can read:

*[...] Due to the limited spatial extent of the VHR dataset, **we used raw Sentinel-1 wrapped interferograms for validate L7/8 surface displacement products (cf. Section 4.1) as well as for classifying the** inventory of gravitational **mass** movements including rock glaciers and landslides (cf. Section 4.3). [...]*

**SC-13.** L222-223: Please elaborate this processing step in a bit more detail. At this point, it remains unclear why and how exactly the DEM was used to identify the PMAs.

**Response >** All the details are described in "Section 4.2". We made the link with the corresponding section. Additionally, we add some clarifications in Section 4.2 to be more clear how we use the DEM to extract PMAs. Now you can read:

*[...] Then, a medium-resolution DEM was used to identify **P**ersistent **M**oving **A**reas **(PMAs) along the slope direction** within the study region (Section 4.2).[...]*

**SC-14.** L237-240: Can you explain in a sentence how the "per-alignment strategy" applied here works? Also, it is not clear to me what "for the small ones" refers to.

**Response >** We added a sentence regarding the pre-alignment strategy. We removed the words "for the small ones" as they were not useful for the text. Now you can read:

*[...] pre-alignment strategy **(i.e. automatic identification of image features matched in a pair of images used then as tie-points)** before the feature tracking stage. Both softwares presents an adaptive windows matching strategy corresponding to 3x3 for MicMac and 7x7 for ASP **as the smallest window size**. [...]*

**SC-15.** L245-246: I am not sure if I understand this processing step. What exactly does "median surface displacement" refer to? Are the authors subtracting the residual mismatches from the total displacement here, or does this refer to the results of the pre-alignment strategy?

**Response >** Even if pre-alignment strategy was used to minimize georeferencing errors, when computing surface displacements over a whole region, the expected value of displacements on stable areas is often not zero (mainly due to the subpixel correlation and changes on the illumination). So, in order to correct this systematic bias, we center EW and NS displacement maps by compute median value over the entire correlation scene, and subtracting to all pixels. Now you can read:

*[...] displacement maps for all dates. **Median value was computed over the entire EW and NS products computing each one for their specific dates.** [...]*

**SC-16.** L247-248: In a nutshell, what processing step was done here?

**Response >** We added a brief explanation about how to minimize the striping effect. Now you can read:

*[...] For the L7/8 dataset, striping effects **from** sensor inter-band misalignments (Ayoub et al., 2008; Leprince et al., 2008) **were mitigated by subtracting the median value of the stacked profile in the along-stripe direction, considering only stable areas (cf. Section 4.3)**.*

**SC-17.** L255-256: I would suggest filling the bracket with the relevant processes here that might operate in your study area. A shifting river reach would also pass the "constant direction" test applied here but is not driven by gravity.

**Response >** We clarify the sentence by removing the three points within the brackets and by adding the example of shifting rivers. Now you can read:

*[...] for motions driven by gravity (rock glaciers, landslides)* ***or other erosive processes (i.e. shifting rivers, river banks erosion)*** *[...]*

**SC-18.** L264: Please explain in a bit more detail how the definition of "moving areas" (I assume this refers to PMA here?) is done. I reckon this is done for the entire time span and based on the cumulative surface displacement? How was a lower threshold applied in this to discern moving and non-moving areas?

**Response >** Indeed, we refer to PMA rather than moving areas. We added a sentence in the same section to clarify this purpose. Regarding the threshold of surface displacement, no threshold was used. Now you can read:

*[...] The time series of* ***cumulative*** *surface displacements from L7/8 images are then used to automatically extract* ***PMAs. The PMA are a group of connected pixels having a coherent movement in time and following the downslope direction****, as this is expected [...]*

**SC-19.** L273: Maybe quickly explain what processes are subsumed under "landslides" and "other". Based on high resolution imagery presented in GoogleEarth, it would be interesting for the reader to know which processes were identified.

**Response >** We added a clarification about the expected geomorphological sign for each class. Now you can read:

*[...]* ***For instance,*** *landslide* ***class was assigned when cracks and scarps were present at the surface and*** *rock glacier* ***class, when typical morphology i.e. front and lateral margins with ridge-and-furrow surface topography was observed.*** *When no clear interpretation about the movement and the geomorphology* ***interpretation*** *could be assessed on either InSAR or Google-Earth basemaps, [...]*

**SC-20.** Table 1 / L295-296: It would be very good to extend this table and add columns that list the number of features that remain after the additional processing steps (i.e. the surface threshold). In the current form, the data presented in L298-320 is a bit confusing, as the numbers mentioned there are not appearing in the Table 1. Furthermore, column one does not extend to the last row, suggesting that "other" is not representing valid data?

**Response >** We expanded the table by adding simple PMA statistics, above the surface threshold (2250 m² - 10 pixels) for each binary class as well as each geomorphological class. In addition, we clarify the paragraph after the table regarding the manual and automatic characterisation. Now you can read:

| TOTAL POLYGONS | Manual characterization | | Above automatic surface threshold (2250 m² - 10 pixels) | |
|---|---|---|---|---|
| | **n** | % | **n** | % |

| Corroborated class | Geomoph class | 1710 | 100 | 975 | 100 |
|---|---|---|---|---|---|
| **NOT CONFIRMED** by InSAR | Sub total | 1209 | 71 | 593 | 61 |
| | Other | 747 | 62 | 382 | 64 |
| | valley bottom | 159 | 13 | 77 | 13 |
| | ridges | 155 | 13 | 79 | 13 |
| | landslide | 17 | 1 | 14 | 2 |
| | rock glacier | 15 | 1 | 5 | 1 |
| | glaciers | 116 | 10 | 77 | 13 |
| **CONFIRMED** by InSAR | Sub total | 501 | 29 | 382 | 39 |
| | rock glacier | 211 | 42 | 153 | 40 |
| | Landslide | 160 | 32 | 105 | 27 |
| | Other | 130 | 26 | 124 | 32 |

*[...] **During the manual characterization process,** we have noticed the presence of an important number of small and isolated polygons **within the 'not confirmed' class** (Fig. 4), **often** located close to the mountain ridges and at the valley bottom (Fig. S3). As these tiny polygons cannot be correctly interpreted, **we set up a surface threshold of 2250 m² (i.e. 10 pixels), to remove them automatically.** [...]*

**SC-21.** L304-306: These numbers for example are not contained in Table 1. Also, the class "deglaciated" is not mentioned there.

**SC-22.** L307-308: I recommend rephrasing this sentence to increase the clarity of what the "automatic surface threshold" is and how this translates into the PMA statistics? It refers to the 10 pixel threshold, right? But how is this automatic? And what does "the most smaller" refer to?

**Response >** Here we answer specific comments 21 and 22. We change the glaciated or glaciated class by 'glacier' to avoid confusion. We check the consistency though the manuscript. In addition, we modify the paragraph to clarify the use of the automatic filter. Now you can read:

*[...] The **selected** surface threshold **seems to be a good compromise to remove noisy (smaller) PMA and keep coherent (larger) PMAs**, by only compromising 15% of **confirmed** PMA (Fig. 4). After the surface threshold is applied **and PMA corresponding to glacier class were removed**, the remaining filtered dataset [...]*

**SC-23.** Figure 4 / L322-325: Maybe show removed PMAs in light colours? In the caption, mention the threshold specifically, instead of "at left of".

**Response >** We think that the black line fits well within the graph. However, we precise the caption with reviewer arguments. Now you can read:

*[...] i.e. 2250 m² (10 pixels) used as a filter to remove **smaller PMAs**. All polygons **below the surface** threshold were removed. [...]*

**SC-24.** L333-334: This is not only related to the technical approach, but also has a morphological cause (i.e. increased friction and low/no ice content in lateral margins) that deserves to be mentioned here.

**Response >** We agree with the reviewer. As the way of how average surface velocities were computed is more a methodological approach, we changed the positions of this paragraph on the new **"Section 4.4 Average spatial velocity and relative changes"**. Within the new section, we added the suggested text:

*[...] As shown in Fig. 3a to d, the pixels located in the borders often have values close to 0 m a⁻¹, due mainly to natural geomorphological causes (i.e. increased friction and low/no ice content in lateral margins) as well as to window sizes of feature-tracking algorithms. [...]*

**SC-25.** L335: This relates to one of my general comments above. I suggest you define this more clearly and use a clear and recognisable terminology throughout the manuscript. E.g. in the next sentences, (L342) does "average velocity" refer to the "Top 50% average velocity"?

**Response >** We removed the fattest pixels and keeped only "Top 50% average velocity". and check the consistency throughout the manuscript.

**SC-26.** Figure 5 / L357-365: If I understand the figure correctly, the viridis colour bar is also valid for the small rock glaciers, but the label "Top 50% mean velocity" is not. In the plots showing the cumulative displacement, error bars would help placing the data into the broader context.

**Response >** You are right about the velocity. We updated the figure legend to clarify the use of the colorbar as well as some clarification about the subplots. In addition, we added the cumulative error bars as well as the names for each study site in bold. Now you can read:

[Figure]

*[...] Figure 5: Surface kinematic characterisation for all PMAs in the central Andes region. a) Illustrates the spatial distribution of all* **confirmed** *PMAs (rock glaciers = 146; landslides = 115; others = 103) coloured by the 'Top 50% average velocity' (viridis colorbar) within the PMA surface. The size of the circle is proportional to the rescaled PMA surface in m²/1,000 for better visualisation. The red letters correspond to the study cases presented in the following subplots. The remaining subplots b) to g)* **(with a suffix of \*.1)** *illustrate mean annual velocity field over the 24 years (2000-2024) for a specific landform* **(name is displayed in bold) where the magnitude of velocity is coloured using viridis colorbar from panel a). Subplots with a suffix of \*.2 represents the** *cumulative surface displacement time series in metres, extracted on the black (and red) point within the landform.* **Cumulative error bars were computed NMAD on stable areas for each date respectively (Section 5.4).***[...]*

**SC-27.** Figure 6 / L404-407: In the panels a) and b), a label indicating which rock glacier is the Tapado and which is the Las Tolas would help the reader following the specific results and discussion sections. Alternatively, this could also be indicated in Figure 5. Furthermore, please elaborate what a "pseudo-GCP" is and how it was constructed – it is not mentioned anywhere else in the manuscript.

**Response >** We updated Figure 6 and added the names for each rock glacier to avoid confusion.

[Figure]

Regarding the pseudo-GCP, this was already mentioned in Section 3.4 GNSS data. We clarify the sentence by adding the number of pseudo-GCPs used in this study and how they were built. Now you can read:

*[...] Additionally, as no GCPs exist for Largo rock glacier, we manually tracked* **13 pseudo-GCPs** *control points on representative features clearly identified on the VHR dataset to compare with L7/8 dataset (cf. Section 5.3). [...]*

**SC-28.** Figure 7 / L419-420: Please elaborate more precisely what is shown in the figure, especially regarding the time span of the averaged data that is plotted here. What exactly does the point to pixel comparison contain? Are these velocities averaged over the temporal range of the entire analysis? In section 3.4 it is mentioned that DGNSS records started in 2009, but what time frame is the comparison with L7/8 and VHR data based on here?

**Response >** The comparison shows average velocity field for GNSS, pseudo-GCPs measurements and average surface velocity fields from remote sensing data. We computed surface velocities using the commission time span from 2009 until 2020. We modify the caption of Figure 7 by adding all these elements. Now you can read:

*[...] Figure 7: Comparison between* **GNSS, pseudo-GCPs** *points* **average velocity** *and* **average surface** *velocity fields from inversion time series for both* **L7/8 and VHR datasets in**

*the subregions of Tapado complex and Largo rock glacier. **The average surface velocities from GNSS measurements, L7/8 and VHR datasets, were calculated according to the common time period, spanning from 2009 to 2020.** [...]*

**SC-29.** L433-435: Again, I don't think this statement is valid here, as the Landsat 7/8 data exploited here also roughly covers the same time span.

**Response >** We agree with the reviewer. However, what we tried to say is that with Landsat imagery you can go back to the mid-1980's. So, remote sensing observations using medium resolution satellite i.e. Landsat covers almost 40 years. We added this statement as well as their respective references to clarify the sentence. Now you can read:

*[...] **but these** datasets are relatively **recent**, covering **only the past 20 years** (Toth & Jóźków, 2016). **In contrast, Landsat imagery (e.g. L4-5-7 or L8) extends back to the mid-1980s (Kooistra et al., 2024; Ustin and Middleton, 2021). However, VHR and freely available SAR** datasets are often limited [...]*

**SC-30.** L449-451: What is the detection threshold mentioned here and how was this determined?

**Response >** The detection threshold mentioned here is the same identified in Section 5.1 (i.e. 2250 m² - 10 pixels). The explanation about how it was defined is explained in Section 5.1 where we mentioned that only a group of contiguous pixels greater than 10 pixels can be interpreted correctly. We added the explanation to clarify the paragraph. Now you can read:

*[...] This **suitability largely depends** on **rock glacier size, and** pixel **coverage within** the landform (Section 5.1). **Here, a minimum surface threshold of 2250 m² (10 pixels) proves effective** for the Andes, but perhaps less so for other **regions with smaller rock glaciers as the** European Alps, [...]*

**SC-31.** L512-515: While I agree that landslides in general might show a higher motion variability, I am not sure that comparing the different processes here helps the line of argument of the manuscript. The term landslides encompasses a range of different gravitational processes and with that a range of movement mechanisms that operate on very different timescales, often rather instantaneous. Also see my general comment on this.

**Response >** Sorry for the misunderstanding. WE agree that landslides encompass a range of different processes. However, we are not comparing processes in the paragraph. The goal of the paragraph is to show that rock glaciers have been better detected than landslides due to their nature (constant viscous flow with lower motion variability compared to landslides. Now you can read:

*[...] Rock glaciers are slightly better detected than other features like landslides, **likely due** to the **lower** motion variability over time. Indeed, rock glaciers are viscous flows (Haeberli et al., 2006) that face changes of **activity** over long periods of time (**Kellerer-Pirklbauer et al., 2022;** Lehmann et al., 2021). On the contrary, landslides can be influenced by seasonal and transient patterns (Lacroix et al., 2020b). [...]*

**SC-32.** Figure 9 / L557-559: It should be made clear what "mean average velocity" means exactly and how this compares with the mean velocity (panel a) and average velocity (panel b). Further, use same unit m yr-1 also here.

**Response >** We modified the X and Y axis titles of Figure 9 to state clearly which statistics has been used. We also change the unit label in the X and Y axis. Figure 9 looks as follows:

[Figure]

Figure **10**: a) Comparison between average velocity computed using the entire PMA surface and 'Top 50%, 30% and 10% average velocity' within PMA. Subplots b), c), and d) show the difference of average velocity 'Top 50%, 30% and 10% average velocity' in respect to the **average** velocity computed over the entire PMA surface.

**SC-33.** L568-569: I might have missed this before, but are these different types of rock glaciers? If so, quickly explain in which respect these are different.

**Response >** The sentence has been removed as this is not coherent with the findings.

**SC-34.** L600-605: If I understand this correctly, it is not only the number of images that is different between both epochs, but also the length of the epochs differ – 14 years (with six images) versus 11 years (with 12 images).

**Response >** This is correct. We added the clarification within the manuscript. Now you can read:

*[...]* ***Observation discrepancies:*** *The 2000 to 2014* ***period includes*** *only six years of observations due to a gap between 2003 and 2013,* ***whereas*** *2013 to 2024* ***has 11*** *years of continuous observations. This may* ***bias*** *the average velocity for each period,* ***conditioning*** *related uncertainties (Fig. 5). The use of ASTER* ***images or other medium-resolution data*** *could* ***help*** *to fill this gap despite the low radiometric resolution.[...]*

**SC-35.** L639-641: Frankly, this seems a bit speculative, and I am not sure which data presented here supports this statement.

**Response >** We modified the sentence to avoid speculative statements. Now you an red:

*[...] **In contrast,** landslides and other **features** are predominant on North-west to East slopes (Fig. 11d). Similar **findings from Blothë et al., (2021)** in the Cordon del Plata **also emphasize** slope orientation as a controlling factor. [...]*

**SC-36.** L660-664: The topographic parameters are not only not covering the entire rock glacier but leave out the feeder basin that delivers material and water to the rock glacier.

**Response >** We added the text as suggested by the reviewer. Now you can read:

*[...] **Although the objective of our study is the monitoring of rock glaciers on a regional scale, other PMAs corresponding to mass movements such as landslides could also be identified. Our results suggest possible correlations between gravitational movements in high mountain areas (e.g. Haeberli et al., 2017; Patton et al., 2019) and permafrost degradation (i.e. freeze, thaw of permafrost) in recent deglaciated areas (Pánek et al., 2022). This study allows us to complete those existing mass movement inventories in the region (e.g. Iribarren Anacona et al., 2015), highlighting areas for further research. While these findings suggest useful regional** relationships between surface kinematics and topographic parameters at the regional scale, **they must be interpreted cautiously. The** morphological statistics **here are derived only** within PMA **boundaries and may not fully represent** the whole **landform** (Fig. S12). **Additionally, PMAs exclude feeder basins, responsible for material supply and water to the rock glacier (Blöthe et al., 2021; Cusicanqui et al., 2021).** Further studies should be conducted to **look at the influences of feeder basins on surface** kinematics **of rock glaciers.** [...]*

**SC-37.** L676: Please explain how this topographical context was extracted from which data set.

**Response >** As the lines 676 does not refer to topographical context, we believe that the referee refers to the caption in Figure 10. We added an explanation to clarify how the topographical context was obtained. Now you can read:

*Figure 1**1**: [...] geomorphological class vs **regional** topographical context **(computed using average pixel frequency from TanDEM-X 12 m DEM)**. a) PMA mean slope; b) d [ ...]*

**Technical corrections**

**SC-38.** L22: "debris frozen landforms" ?

**Response >** Modified as suggested.

**SC-39.** L26: word missing or unclear: "10% of the accelerations in one decade"?

**Response >** Modified as suggested. Now you can read:

*[...] **Nevertheless,** decadal velocity changes **were observed in 3% of PMAs, where three** rock glaciers **showing 11% increase and six rock glaciers showing an 18% decrease in velocity over one decade**. [ ...]*

**SC-40.** L32: Please rephrase! Do you mean "snow cover variability"?

**Response >** Modified as suggested.

**SC-41.** L70: delete "and"

**Response >** Modified as suggested.

**SC-42.** L71: a lot of "thus" here

**Response >** Modified as suggested.

**SC-43.** L86: "set tracking techniques"?

**Response >** We change "set" by "feature" as suggested.

**SC-44.** L94: If I remember correctly, Eriksen et al. 2018 investigate a rock glacier in Norway…

**Response >** We remove the reference. Modified as suggested.

**SC-45.** L105: at not "ate"

**Response >** Modified as suggested.

**SC-46.** L106: explain "VHR" here

**Response >** Modified as suggested.

**SC-47.** L124: have shown a warming trend?

**Response >** Modified as suggested.

**SC-48.** Figure 2: the font of the panel labels as well as the text is not consistent throughout the figure.

**Response >** We modified the font of panel b as suggested.

**SC-49.** L203 & 216: L7/8

**Response >** Modified as suggested.

**SC-50.** L285: Please provide a more appropriate heading here

**Response >** Modified as suggested. Now you can read:
*[...] 5.1 Characterization of PMA's extraction [...]*

**SC-51.** L296: "is presented in" instead of "could be found"

**Response >** Modified as suggested.

**SC-52.** L311: This should be 58%, right?

**Response >** Modified as suggested.

**SC-53.** L332: "taking into account"

**Response >** Modified as suggested.

**SC-54.** L366: Although

**Response >** Modified as suggested.

**SC-55.** L387: rock glaciers and landslides instead of 'rock glaciers' and 'landslides', or alter in the remaining manuscript

**Response >** Modified as suggested.

**SC-56.** L390: delete "further", as this is the results section

**Response >** Modified as suggested.

**SC-57.** L417: I am not sure if "were drawn" paints the correct picture here

**Response >** We replace"were drawn" by "are located". Modified as suggested.

**SC-58.** L435: Elaborate which datasets you are referring to here.

**Response >** We replaced by "On the other hand, VHR and InSAR". Modified as suggested.

**SC-59.** L437: "with a global reach"?

**Response >** We replaced by "with a global coverage" Modified as suggested.

**SC-60.** L446: Halla et al. 2021

**Response >** Modified as suggested.

**SC-61.** L458, 460, 475: I think the text here refers to Fig. 6, not 7?

**Response >** Modified as suggested.

**SC-62.** L523: Explain the abbreviation RoGI, also use consistently (L525, Rogi)

**Response >** We replace "RoGI" by "rock glacier inventory". Modified as suggested.

**SC-63.** L568: underestimation

**Response >** Modified as suggested.

**SC-64.** L630,635: I think the text here refers to Fig. 10, not 11?

**Response >** Modified as suggested.

**SC-65.** L646: a lot of permafrost here

**Response >** Modified as suggested.

**SC-66.** L671: Conclusions

**Response >** Modified as suggested.

**SC-67.** L715-716: repetition

**Response >** Modified as suggested.

---

## Author Comment (AC2)

**Response to comments by second reviewer – Anonymous Referee**

We sincerely thank the Anonymous referee for the thoughtful comments and constructive suggestions, which have greatly contributed to improving the clarity and overall quality of our manuscript. Below, we provide a point-by-point response to each comment.

**General comments**

**GC-1.** Cusicanqui et al. estimate rock glacier kinematics on annual to decadal time scales from medium-resolution Landsat imagery. Assessing the applicability of Landsat imagery for this purpose is important because Landsat images are more widely available than higher-resolution images, while the lower spatial resolution of 15 m (panchromatic) raises questions about the suitability for measuring rock glacier kinematics on subdecadal time scales. To appraise the applicability, the authors compare the Landsat-derived motion estimates to independent observations derived from GNSS and high-resolution images, as well as to an InSAR inventory. The study raises and addresses a question of substantial interest to readers of The Cryosphere. Its novelty lies in its being the first to use Landsat imagery for estimating rock glacier kinematics. The methods and data interpretation are, for the most part, sound, and the results are valuable to the community.

**Response >** We thank the reviewer for the positive comments about our study.

**GC-2.** While the manuscript clearly advances the field, I have concerns about the extent to which the presented data support the authors' conclusions about the observational uncertainty. Furthermore, I have found the manuscript difficult to follow because of inconsistencies in content and terminology and, at times, a writing style I consider to be verbose and vague. As I agree with Jan Henrik Blöthe's three general comments, I will not comment further on these aspects, focusing instead on my concerns about the uncertainty analysis and the presentation.
Uncertainty: My main content-related concern pertains to the accuracy assessment. There is currently no figure or table that shows aggregated accuracy metrics, making it difficult for the reader to appraise the evidence for claims found in the abstract and conclusion. I believe it would be helpful to reorganize the accuracy assessment, introducing relevant equations in the methods and presenting the estimates in the results (including a figure with metrics such as the root mean square deviation with respect to independent estimates, the NMAD over stable areas, or the estimated deviation between temporal changes). In the abstract and conclusions, these specific metrics can be reported, while clearly distinguishing observations from subjective interpretation.

**Response >** We thank the reviewer for highlighting this point. Regarding the uncertainty, as suggested by the reviewer, we have reorganized the manuscript by adding a specific section showing the reported uncertainties; in this Section 5.4, we have now added a new figure showing the variation in time of NMAD on both components EW and NS, as well as Table

showing all the statistics for both L7/8 and VHR datasets, on stable and mobile areas (i.e. differences with GNSS observations). Now you can read:

*[...]*

*5.4 Reported uncertainties*

[revised manuscript text omitted]

**Response >** As explained in Section 4.4. (Equation 2) and Section 5.5. (see the text below), the uncertainty of velocity change depends on the magnitude of the velocity of both encountered periods, for instance 2000-2013 and 2013-2014. An increase in velocity of 100% on small velocity magnitudes encompasses higher uncertainties. This is not the case with higher velocity magnitudes. The main reason behind it is the high uncertainties intrinsic to the L7/8 dataset and the error propagation through time. Please refer to Section 5.5 (text copied below) for more details about the given examples.

*[...]*

**5.5 Velocity changes**

*Using 24-year surface displacement data, decadal velocity changes were analyzed by calculating surface velocities over two periods: 2000–2014 ($V_1$) and 2013–2024 ($V_2$), for all PMAs. The uncertainty in velocity change depends on the magnitude of velocity in both periods (Eq. 2). Smaller velocity magnitudes result in greater relative uncertainties, whereas higher velocities yield proportionally smaller uncertainties. **To illustrate, a velocity increase from 0.5 to 1.0 m $a^{-1}$ (100% change) has an uncertainty of 0.78 m $a^{-1}$, representing 78% of the relative change. In contrast, an increase from 1.0 to 2.0 m $a^{-1}$ has an uncertainty of 0.39 m $a^{-1}$, or 39% of the relative velocity change. Consequently, only PMAs with velocities exceeding 1 m $a^{-1}$ can be considered reliable for statistically significant velocity change.** By considering PMAs with velocities exceeding 1 m a-1, nine 'rock glaciers' and 2 'landslides'. Within those selected features, three rock glaciers showed an increase in velocity 11% (Fig. 5c), whereas six rock glaciers showed a decrease in velocity of 18% over two decades. On the other hand, two Landslides exhibited a larger increase in velocity of 50%. For further discussion, please refer to Section 6.5.*
*[...]*

**GC-4.** Was the accuracy of decadal velocity changes evaluated directly based on quantitative data? Section 5.2 contains a back-of-the envelope appraisal of the expected uncertainty in the relative change (which would fit better in the methods), but a dedicated assessment is missing. Furthermore, equation (2) seems suspect, as the numerator can be negative (I suppose it should be replaced by the geometric mean). The assumptions should be made explicit: If two quantities are assumed to be equal, it is not sufficient to say they are assumed to be similar. Section 5.3 contains an evaluation of velocity estimates, not of changes in velocity. The statement (and Section 5.2) mentions statistical significance, but it is not clear to me how and under what assumptions statistical significance was determined.

**Response >** Regarding the uncertainties assessment and the suggestion of moving the Section 5.2 in the methods section, please refer to the response of **GC-2** in this document to look more in detail about uncertainties assessment and the **new "Section 4.4. Average spatial velocity and relative velocity changes"** in the method section.

Regarding the Equation 2, the numerator can not be negative because there is no negative velocity. If the reviewer refers to the Equation 1, the value of velocity change could be negative because we are computing relative velocity changes in respect to the first period. Negative velocity changes reflect decelerations, compared to the first selected period.

$$V_{change} = \frac{V_2 - V_1}{V_1} \tag{1}$$

$$\sigma V_{change} = \left( \frac{V_2 + V_1}{V_1^2} \right) * \sigma V , \tag{2}$$

**GC-5.** In the statement from the conclusion, the observed underestimation is attributed to discrepant spatial scales and temporal gaps. It is not apparent to me what dedicated quantitative analyses were conducted to establish this conclusion, or whether it is a subjective interpretation.

**Response >** The observed underestimation of surface velocities was evidenced when the comparison between GNSS and pseudo-GCPs and satellite image correlation was carried out. In fact, this underestimation is highlighted in "Section 5.3 Velocity validation using GNSS and VHR datasets", from where, a quantitative analysis has been dedicated to this (more specifically in Figure 6 and 7). We added a numerical comparison between GNSS and pseudo-GCPs in the new Table 2 (please refer to the response of **GC-2** in this document) in the **new section 5.4. Reported uncertainties**.

**GC-6.** Presentation: I have found the manuscript difficult to follow, primarily for two reasons:
1) Cohesion: Discrepancies in content and terminology between the different sections presented challenges to my understanding of the manuscript. I was repeatedly taken by surprise by sudden changes in direction: The results introduce new methods and analyses that were not covered in the methods, while the discussion introduces new results and analyses not mentioned previously. The introduction contains a long literature review, but I have found it difficult to relate it to the remainder of the manuscript. In particular, the discussion section comprises six subsections, of which at least three have no easily discernible (for me) connection to the introduction or conclusion:

**Response >** We have corrected and homogenized all existing discrepancies in the terminology.

Regarding the "new methods" found in the discussion sections, we agree on the fact that some paragraphs were better placed in the methods and results sections. In this sense, we move those paragraphs to the corresponding sections in methods (**new Section 4.4 Average spatial velocity and relative velocity changes**) as well as one subsection from the discussions.

Regarding the introduction, we think that the introduction summarizes all key points related to how surface velocity was estimated classically, as well as all the limitations regarding each methodology (i.e; InSAR and optical imagery). We believe that all these statements are necessary to better know the reader, what is the main contribution of our robust methodological approach.

**GC-7.-** Section 6.2: I am not sure how this discussion relates to the objectives of the manuscript, as its connection to the presented results is tenuous. A new figure is introduced, but it is not described in detail. Is the primary purpose of the InSAR to classify the PMAs (together with Google Earth imagery) or does it also contribute to the quantitative appraisal of the Landsat results?

**Response >** The primary goal of InSAR wrapped interferograms was to corroborate PMAs. As PMAs come from an automatic extraction, it is not except for errors. This is evidenced in Table 1 showing that 60% of PMA were not corroborated by InSAR (mostly the smallest ones). The goal of the Figure 8 is to show the good agreement of largest PMAs as well as the ambiguity of some other features, notably of landslides, who may have been active during several years and were not detected within L7/8 dataset (Fig 8d and e) and thus, they don't appear on wrapped interferogram. Clear fringe patterns are visible on 60 days of interferograms, and not identified at all with L7/8 (Fig. 8b), very likely related to the type of landform (very slow moving landslides and slow rock glaciers). In addition, PMAs are often located in the middle and frontal sector of the landform, with some discrepancies in the upper region (Fig. 8d).

**GC-8.-** Section 6.3: Consider moving the new results shown here to the results section. In addition, the more tightly this analysis is integrated with the remainder of the manuscript, the easier it will be for the reader to appreciate it.

**Response >** We moved the entire paragraph as suggested. Please refer to the response of **GC-2** for a look at the changes..

**GC-9.-** Section 6.6: Introduces new results (Fig 10) that are not referred to elsewhere in the main body of the manuscript. Consider cutting it or motivating it.

**Response >** We decided to motivate this section rather than cutting it. We agree that this is not the main focus of the current manuscript. However, not so much studies have focused on regional patterns. We find it a bit unfortunate not to be able to value this regional aspect. So we decided to motivate the section. Now you can read:

*6.5 **Wider geomorphic implications** of PMA*

*Understanding the broader geomorphic implications of PMAs is critical for interpreting their role within high mountain environments and their response to climatic and*

*geomorphological processes. While much of the manuscript focuses on kinematic and spatial characteristics of PMAs, this section aims to contextualize the observed patterns within a regional framework, bridging findings with topographic and geomorphological contexts, shedding light on the factors influencing their spatial distribution and surface dynamics. The PMAs in the study area show heterogeneous spatial distribution across topographic conditions (Fig. 5a). Analysis of the Top 50% average velocity and its relationship to slope, aspect, elevation and surface area, derived from the TanDEM-X 12.5 m DEM, reveals several key patterns (Fig. 11, Fig. S7)*

**GC-10.** Inconsistency in terminology also presents challenges to the reader. For instance, the comparison to GNSS is referred to as "ground truth" in the results only, while the word GNSS is only used in the subsection header in the results. Furthermore, the expression "false PMA" is only used in the discussion.

**Response >** We agree on the inconsistencies regarding "ground truth" terminology. We checked all the inconsistencies throughout the manuscript and we were replaced by GNSS.

**GC-11.** 2) Style: I find that the verbose style detracts from the content of the manuscript. I believe that reducing the word count by 25% is a realistic target. Removal of filler phrases such as "we can also state that", "we proceeded to compare surface velocity fields in more detail", "as mentioned in", "on the other hand", or "briefly", would help the reader focus on the content. So would strong topic sentences that succinctly summarize the content of the paragraph, thus guiding the reader through the manuscript. I provide more specifics in the minor comments.

**Response >** We effectively reduced 15% of the text guaranteeing the coherence and substance of the message we want to convey. In addition, we removed those "filler phrases" to avoid redundancy within the manuscript.

**GC-12.** In addition, extensive language edits are advised, as illustrated by the following phrases from the abstract: "The results of this study shows [...]" -> show; "over a 24-years" -> over a 24-year period or over 24 years; "of which 153 corresponds" -> correspond; "providing an alternative to InSAR, for monitoring": remove comma.

**Response >** Modified as suggested.

**Specific comments**

**SC-1.** Title: Kinematics?

**Response >** Modified as suggested.

**SC-2.** Abstract: Mention the study area?

**Response >** We added some text in the abstract. Now you can read:

*[...] (153 corresponding to rock glaciers) **over a 24-years, over an area of 2250 km².** [...]*

**SC-3.** l23: Isn't it the method that underestimates the velocity?

**Response >** Not necessarily. As mentioned in Section 4.1, we applied the same methodology to two different types of data i.e. L7/8 and VHR. The results shown in Fig. 6 and Fig. 7, do not show underestimation on the VHR dataset. Only L7/8 underestimate the velocity. Mainly due to the coarse pixel size (i.e. 15 m) of L7/8 plays an important role on well discriminating detailed surface velocity patterns, resulting in an underestimation of velocities. All these aspects have been discussed in Section 6.1.

**SC-4.**  l115: rapid: rugged or steep?

**Response >** We change "rapid" by "rugged" as suggested. Now you can read:

*[...] The **rugged** topography from coastal position [...]*

**SC-5.**  l129-130: Incomplete sentence? Consider cutting the entire paragraph.

**Response >** We precise the sentence. No you can read:

*[...] More recent studies highlight the complex interaction between remnants of glaciers, debris covered glaciers and rock glaciers (Navarro et al., 2023a; Robson et al., 2021) as well as the importance of rock glaciers as water storage resources (MacDonell et al., 2022; Schaffer et al., 2019; Schaffer and MacDonell, 2022). [...]*

**SC-6.**  l160: "data gap existed": Spatial data gaps due to SLC failure or data gap because you did not include these images?

**Response >** We did not include images within the 2004 - 2013 period for both reasons. We clarify this statement in the text. Now you can read:

*[...] **However, due to the** Scan Line Corrector **failure** on the **Landsat-7** satellite **between 2004 and 2013** (Markham et al., 2004), **we excluded Landsat-7 scenes within this period to avoid data gaps.** [...]*

**SC-7.**  l196: I do not know what you mean by "interferograms averaged in 2-looks", as subsequently a 2 by 8 boxcar filter is mentioned.

**Response >** Multilooking processing is an optional step on SAR processing and is used to produce a product with nominal image pixel size. Multiple looks may be generated by averaging over range and/or azimuth resolution cells improving radiometric resolution but degrading spatial resolution. As a result, the image will have less noise and approximate square pixel spacing after being converted from slant range to ground range. The text refers briefly to the methodology used for multilook FLATSIM interferograms to advertise to the reader how the interferograms were built.

**SC-8.**  l236: "The MGM algorithm reduces the amount of high-frequency artefacts": Compared to what method? High-frequency: what (presumably spatial) frequencies are being referred to?

**Response >** We added a clarification to the sentence. Now you can read:

*[...] The MGM algorithm reduces the amount of high-frequency **spatial** artefacts **(compared to classic Block-Matching algorithms)** in textureless regions and produces smooth surface displacement fields. [...]*

**SC-9.**   l249: "redundant system" of equations? It would help to be explicit on the assumptions and methods here. Any regularization?

**Response >** No, is the redundant system of pixels. No regularization is used in the redundant system. Please refer to Bontemps et al., 2018 for all the details.

**SC-10.**   l260: Is the slope direction oriented down or up the slope?

**Response >** is the downslope direction. We added this clarifiction on the text.

**SC-11.**   l285: Fix subsection header

**Response >** changed as suggested.

**SC-12.**   Table 1: Consider changing the class names "non valid" (or invalid?) and "valid" to something like not corroborated by / corroborated by InSAR to better convey the substantial epistemic uncertainty

**Response >** As suggested, we change the class names "non valid" and "valid" by "confirmed by InSAR" and "not confirmed by InSAR". We check the consistency throughout the manuscript.

**SC-13.**   l305: of all PMAs?

**Response >** changed as suggested.

**SC-14.**   l330: How was the NMAD computed (equation)? What normalization was applied?

**Response >** As Normalized Mean Absolute deviation (NMAD) is a common statistical metric to assess uncertainties in geospatial sciences, we belive that introduce the equation in the manuscript wont add any useful information. Nevertheless, we refer the reader to the scientific publication about NMAD meaning. Now you an read:

*[...] The Normalised Mean Absolute Deviation (NMAD; **Höhle and Höhle, 2009**) computed over stable [...]*

**SC-15.**   l334: "due to window sizes of feature-tracking algorithms": Is this the only conceivable reason? If the attribution is speculative, consider removing it from the Results.

**Response >** We added some clarification why borders has velocities close to 0. Now you can read:

*[...] Fig. 5 b to g, the pixels located in the borders often have values close to 0 m a$^{-1}$, due **mainly to natural geomorphological causes (i.e. increased friction and low/no ice content in lateral margins) as well as** to window sizes of feature-tracking algorithms. [...]*

**SC-16.**   l377: "It can be observed that": where?

**Response >** We removed the sentence as suggested.

**SC-17.** l392: A concise topic sentence that summarizes the entire paragraph would make this paragraph easier to read.

**Response >** Changed as suggested. Now you can read:

*[...] We compare surface velocity fields in more detail for the two selected sub-regions around the Tapado and Largo rock glaciers (Fig. 2a). [...]*

**SC-18.** l395: What is the time period of the various estimates? Where changes in displacement rates evaluated?

**Response >** Those details are already explained in Section 3.4 as well as Figure 6. Please Please refer to Figure 6 for more details.

**SC-19.** l401: important != big

**Response >** We prefer to keep important instead of big because even if both surface velocity fields are different, the underestimation of velocity goes up 30% only.

**SC-20.** l409: Is this the root mean square deviation?

**Response >** The value corresponds to the standard deviation of the comparison.

**SC-21.** l449: I am not sure what the hypothesis is, why it is rough, and its precise relation to the rest of the sentence.

**Response >** The sentence talks about the suitability of L7/8 for monitoring rock glaciers across the world. We modified the sentence to remove the ambiguity. BNow you can read:

*[...] This **suitability largely depends** on **rock glacier size, and** pixel cover**age within** the landform **(Section 5.1)**. [...]*

**SC-22.** l464: what do you mean by "enhance/difficult"

**Response >** When using remote sensing imagery, the presence of ridges and furrows can sometimes enhance the contrast and texture of the surface. On the other hand, when images are taken with different solar angles, the shadow effect on ridges and furrows structure could also make feature tracking processing difficult. This is something well known on feature tracking algorithms and well explained in Kääb & Heid, (2012).

**SC-23.** l568: sub-estimation -> underestimation?

**Response >** Changed as suggested

**SC-24.** l578: "NMAD over stable areas corresponds to 60% of the [area]" What do you mean?

**Response >** The entire surface considered as a stable area used to quantify NMAD values is equal to 60% of the entire area. We removed this sentence from here as it is to far for the reader

to discover the percentage of the areas used to compute NMAD. We placed the correctes sentence in Section 5.2. Now you can read:

*[...]* Our calculations indicate that the NMAD over stable areas (Fig. S4) equates to 0.07 m a$^{-1}$ ± 0.16 as standard deviation. *[...]*

**SC-25.** l671: -> Conclusion

**Response >** Changed as suggested

---

## Referee Report (RR1)

I am grateful to the authors for the detailed response and the various improvements to the manuscript. In particular, the method description has improved considerably, and the added subsection on uncertainty estimates in the results also strengthens the manuscript.

However, I continue to have major concerns about the uncertainty analysis of the velocity change and the evidential basis of several statements in the conclusion section. In addition, the presentation warrants further improvement, as language issues such as incomplete sentences detract from the content.

**1) Velocity change analysis** I continue to have major reservations about the correctness of Equation 2 and am concerned about the absence of quantitative estimates and uncertainty estimates in Section 5.5.

a) Equation 2

Equation 2 looks wrong and the assumptions and derivation remain unclear. Assuming the errors in $V_1$ and $V_2$ are uncorrelated and their variances identical and equal to $\sigma_V^2$ (none of these assumptions is invoked in the manuscript), the variance of $f(V_1, V_2) = (V_2 - V_1)/V_1$ is, to leading order, given by

$$
\begin{aligned}
\sigma_f^2 &= \sigma_V^2 \left( \frac{\partial f}{\partial V_1} \right)^2 + \sigma_V^2 \left( \frac{\partial f}{\partial V_2} \right)^2 \\
&= \sigma_V^2 \left( -\frac{V_2}{V_1^2} \right)^2 + \sigma_V^2 \left( \frac{1}{V_1} \right)^2 \\
&= \sigma_V^2 \frac{V_1^2 + V_2^2}{V_1^4},
\end{aligned}
$$

(1)

which upon taking the square root yields a standard error estimate of

$$
\sigma_f = \sigma_V \frac{\sqrt{V_1^2 + V_2^2}}{V_1^2}.
$$

Conversely, the authors' equation (2) replaces the numerator with $V_1 + V_2$. I mentioned last time that their equation can give meaningless negative results for negative values of $V_1$ or $V_2$.

b) Results: Section 5.5

The second half of the only paragraph states selected relative velocity changes, but no comprehensive presentation of quantitative estimates or their uncertainty is provided. While Fig. 5 is referenced, neither the relative velocity change nor its uncertainty is directly shown. Given the claims in the conclusion about the estimation of velocity changes, I feel the reader needs to be presented quantitative information to appraise the authors' claims.

**2) Unclear basis for conclusions** The conclusion sections continues to contain statement for which I cannot identify solid evidence. In particular,

a) A sentence I highlighted in the same context in the first round "Although underestimations occurred due to pixel size, temporal data gaps and velocity field heterogeneity, decadal velocity changes were observable under certain conditions, particularly for features exceeding 1 m a-1". What is the evidence that pixel size and temporal data gaps caused the observed underestimation? What is the precise meaning of observable, and how was this conclusion established? I assume you mean features whose velocity exceeds 1 m/a on average.

b) "Below this threshold, velocity changes detected with L7/8 data were not statistically significant." I am not aware of any formal statistical test that was conducted (see 1) and hence cannot identify the empirical basis for this claim.

**3) Presentation** I encourage the authors, especially the senior authors, to make further improvements to the presentation. Language issues include missing articles (e.g. "Average velocity field" in l 296 [or plural]), incomplete sentences ("By considering PMAs with velocities exceeding 1 m a-1, nine rock glaciers and 2 landslides." in line 467); or sentences such as the following from line 312:
"Stable areas were defined using TanDEM-X DEM and slopes lower than 35°, without taking into account neither glacier outlines with a buffer of 500 m for each glacier (RGI Consortium, 2017) nor all PMAs, also not confirmed ones produced in this study."
Do you mean that glaciers and their surroundings (500 m buffer) and PMAs were excluded?

This sentence is a representative example of the verbose style that I find difficult to follow. Adopting a more concise style and reducing the word count by ∼20% could make this manuscript easier to follow.

**Additional remark** Consider including a quantitative statement on the uncertainty estimate in abstract and conclusion.

---

## Author Response (AR2)

**2nd round of review**

**Response to comments by Handling editor Prof. Christian Hauck**

**General comments**

• Thank you again for your author comments and revisions of the manuscript. The reviewers of the first round re-evaluated your manuscript and your changes, and both found that the manuscript improved substantially. However, they both found that there still remain several open questions that were not addressed sufficiently. I would like to especially mention the points 1 and 2 of referee 2, as they address potential errors or unclear conclusions in the manuscript. But also the other points mentioned by the reviewers need to be addressed and improved before publication is possible.

**Response >** We sincerely thank you for your feedback and for coordinating the re-evaluation of our manuscript. We are pleased to learn that both reviewers recognized a substantial improvement in the revised version. We also appreciate the continued critical assessment and fully acknowledge that several aspects still required clarification or further elaboration.

In particular, we have carefully addressed the concerns raised in Points 1 and 2 by Referee 2, as highlighted in your message, by providing additional explanations, data support, and revised interpretations where necessary. Furthermore, we have systematically responded to all remaining comments from both reviewers to ensure that all outstanding issues have been resolved. The main revisions incorporated in this second version of the manuscript include:

- Equation 2 and related calculations have been updated, which required a revision of the section discussing velocity changes (Section 5.5).
- Figures 8 and 9 have been moved to the supplementary material. This decision was based on the fact that these figures were infrequently referenced in the manuscript and provided limited added value for the reader. In contrast, we have added a new Figure 9 (see our response to Referee 2), which better illustrates the velocity changes and associated uncertainties derived from the L7/8 dataset for those PMAs where such changes could be detected.
- In order to be more clear about the validation of PMAs using high resolution GoogleEarth imagery, we replace the 'other' geomorphic class by 'unclassified'. Please refer to Section 5.1.
- We acknowledge the reviewers' comments regarding the writing style and clarity. Although we had already reduced the manuscript length by 15% in the first revision, we conducted another thorough review focused on simplifying complex sentence structures, particularly those in the past tense. This led to an additional 10% reduction in manuscript length, without compromising scientific clarity or coherence. We believe that further reductions would risk omitting essential content. In addition, language and style issues have been addressed throughout the text to ensure greater precision and consistency.

We believe that the revisions made have substantially improved the clarity, coherence, and overall robustness of the manuscript. All changes are clearly highlighted in the revised version, and detailed point-by-point responses to the reviewers' comments are provided below.

**Response to comments by Dr. Jan Henrik Blöthe**

We sincerely thank Dr. Jan Henrik Blöthe for the thoughtful comments and constructive suggestions, which have greatly contributed to improving the clarity and overall quality of our manuscript. Below, we provide a point-by-point response to each comment.

**General comments**

- This is a revised version of the original submission that I commented on earlier. I will quote my summary of the original submission, before mainly focussing on the replies to my earlier comments but also taking into consideration the revised manuscript in general.

"In their manuscript, Cusicanqui et al. use freely available Landsat 7 and 8 imagery to derive velocities of rock glaciers (and other detectable movements) in the Andes of Chile and Argentina over the course of more than two decades. Checking their derived velocities against a complementary data set using InSAR approaches as well as high-resolution data sets and DGNSS measurements for two selected field sites in Chile, the authors demonstrate the feasibility of using medium resolution imagery for the robust quantification of rock glacier surface movement."

In their revised manuscript the authors have adjusted the text and figures largely in line with the comments made by one anonymous reviewer and me. Overall, I am convinced that the manuscript has significantly improved in overall quality. More specifically, the inclusion of a larger section that specifically addresses the uncertainties associated with the analysis has added to the scientific rigour of the work.

Regarding my previous comments and suggestions, I will outline a few specific comments below (Line numbers refer to the document "egusphere-2024-2393-ATC1.pdf") that should in my view be addressed before accepting the manuscript for publication.

Apart from this, I recommend to once again critically go through the manuscript and check for inconsistencies and clarity of writing. Though the clarity has improved, there are still some statements that could be formulated with more precision.

In general, the authors followed most of my suggestions, in some cases more literally than I would have imagined. The authors put a lot of effort into adjusting the figures of their manuscript and this clearly helped to improve the quality of their work. Especially Fig. S15, the adjustments to Fig. 5 and the inclusion of the new Fig. 8 and Tab. 2 are important to mention here.

- Even though the authors adjusted parts of the text, I am not fully convinced that the terms gravitational mass movements, landslides, and rock glaciers have been defined in a sufficient clarity. I commented on this aspect extensively (GC-1, SC-2, SC-12, SC-19), and I acknowledge the decision by the authors to keep the landslide movements in their analysis. However, following a clear and consistent terminology would be desirable for the clarity of the text. In this regard, "gravitational mass movements" (L232, 403; "gravitational slope mass movements" (L22) ◊ is this a typo?) seems a fitting term, though most geomorphology textbooks would not list "rock glaciers" under the term "mass movements". I would therefore recommend to separate the identified landforms, e.g. in L22, into 153 rock glaciers and 229 gravitational mass movements. In the introduction, I want to recommend to include a clear definition of what

is considered a "mass movement" and a "landslide", as these are terms that are not used consistently throughout the manuscript.

**Response >** We agree with the reviewer regarding the inappropriate use of the term "gravitational mass movements," which does not accurately encompass all the landforms identified in our study. This imprecision likely stems from common usage within the landslide research community, where "mass movement" is often employed as a synonym for "landslide." To eliminate this ambiguity, we have added a clear definition of "landslide" in the Introduction. Furthermore, we have removed the term "gravitational mass movement" throughout the manuscript and now explicitly refer to the specific landform types identified. The revised text now reads as follows:

> *[…] (Cusicanqui et al., 2021; Haberkorn et al., 2021). These changes also favour landslides (**the downslope movement of soil, rock, and organic materials under the force of gravity**). **For instance, recent warming induces an increased frequency of landslides (Pei et al., 2023).** However, warming affects mountain permafrost differently […]*

On the other hand, we also corrected term "gravitational mass movements" within the lines L22, L232 and L403. Now you can read:

> *L22: […] quantification of surface kinematics of **1153 rock glaciers, 124 landslides and 105 unclassified landforms** over a 24-years, across a 2250 km² area. […]*

> *L232: […] Due to the limited spatial extent of the VHR dataset, **we used raw Sentinel-1 wrapped interferograms to validate the classification of the L7/8 surface displacement products** (cf. Section 4.3). […]*

> *L403: […] These confirmed PMAs correspond to rock glaciers and mostly large landslides (Tab. 1) […]*

**Specific comments**

**SC-1.** L329-330: Please rephrase this sentence. Geomorphology interpretation = geomorphic interpretation?

**Response >** We modified the sentence to be more specific. Now you can read:

> *[...] When no clear interpretation about the movement and **geomorphic interpretation** could be assessed on either InSAR or Google-Earth basemaps, [...]*

**SC-2.** L348-349: Maybe rephrase and elaborate this slightly, my comment was rather thought as a suggestion, not for literal use;-) In my view, natural behaviour/reasons would be more fitting than causes.

**Response >** We modified the sentence to be more specific. Now you can read:

> *[...] due mainly to natural **behaviour of rock glaciers—**increased friction and low/no ice content in lateral margins—as well as [...]*

**Technical corrections**

**SC-3.** L308: "river banks erosion" ◊ river bank erosion

**Response >** Modified as suggested

**SC-4.** L860: "recent deglaciated" ◊ recently deglaciated

**Response >** Modified as suggested

**SC-5.** L866: better: Material and water supply?

**Response >** Modified as suggested

**SC-6.** I realize that the authors changed the unit from m yr-1 to m a-1 throughout the manuscript. This might have been in reaction to a comment by the anonymous reviewer? Looking at the guide for authors from TC, I would think that m yr-1 is more appropriate:

o "Ma and Myr (also Ga, ka; Gyr, kyr): "Ma" stands for "mega-annum" and literally means millions of years ago, thus referring to a specific time/date in the past as measured from now. In contrast, "Myr" stands for millions of years and is used in reference to duration (CSE, p. 398; North American commission on stratigraphic nomenclature)." ◊ https://www.the-cryosphere.net/submission.html

**Response >** Thank you for your suggestion. We carefully reviewed the International System of Units (SI) Brochure available on the Copernicus website. The document does not explicitly state a preference between the use of $m\ yr^{-1}$ or $m\ a^{-1}$ for expressing velocity. In light of this, and to maintain consistency and clarity, we have standardized all relevant notations in the manuscript and figures using $m\ yr^{-1}$.

**Response to comments by Anonymous Referee**

We sincerely thank the Anonymous referee for the thoughtful comments and constructive suggestions, which have greatly contributed to improving the clarity and overall quality of our manuscript. Below, we provide a point-by-point response to each comment.

**General comments**

**SC-7.** I am grateful to the authors for the detailed response and the various improvements to the manuscript. In particular, the method description has improved considerably, and the added subsection on uncertainty estimates in the results also strengthens the manuscript. However, I continue to have major concerns about the uncertainty analysis of the velocity change and the evidential basis of several statements in the conclusion section. In addition, the presentation warrants further improvement, as language issues such as incomplete sentences detract from the content.

Velocity change analysis I continue to have major reservations about the correctness of Equation 2 and am concerned about the absence of quantitative estimates and uncertainty estimates in Section 5.5. a) Equation 2: Equation 2 looks wrong and the assumptions and derivation remain unclear.

**Response >** We sincerely thank the reviewer for identifying this error. We acknowledge that there was a mistake in the derivation of Equation 1 related to the calculation of relative uncertainty. This has now been corrected, and the updated Equation 2 is presented as follows:

$$\sigma V_{change} = \sigma V \frac{\sqrt{V_1^2 + V_2^2}}{V_1^2} \tag{2}$$

This error led to an underestimation of the relative uncertainty. To illustrate the implications of this correction, we compared the uncertainties in velocity changes obtained using both the incorrect and corrected formulas (Figure A). Figure A displays the percentage of velocity change on the x-axis and the corresponding uncertainty on the y-axis for both formulations. As shown in the figure, the previously incorrect derivation resulted in a consistent overestimation of uncertainties. This overestimation reaches up to 57% for PMAs with smaller velocity magnitudes (0.1–0.3 m yr⁻¹), and decreases significantly for velocities above 0.3 m yr⁻¹. For PMAs with an average velocity of 1 m yr⁻¹, the overestimation between the two formulas is reduced to approximately 6%.

Figure A. Modeling of relative velocity changes (ranging from -100% to 100%; Eq. 1) and their respective uncertainties (Eq. 2) using correct and wrong derivation (respectively) for various velocity magnitudes (0.1 – 1 m yr⁻¹).

Accordingly, we have updated all affected calculations. However, the overall results remain largely unchanged, particularly regarding the number of PMAs for which velocity changes exceed the uncertainty threshold (σv change). The number of PMAs meeting this criterion decreased slightly, from 11 (9 rock glaciers and 2 landslides) to 8. Among these, we identified 3 rock glaciers, 2 landslides, and 3 unclassified PMAs.

We have revised Section 5.5 to reflect these updates and included a new figure to present the results more clearly. The updated Section 5.5 now reads as follows:

[Figure]

Correct formula: $\sigma V_{change} = \sigma V \frac{\sqrt{V_1^2 + V_2^2}}{V_1^2}$

Wrong formula: $\sigma V_{change} = \left(\frac{V_2 + V_1}{V_1^2}\right)\sigma V$

*[…] Using the 24-year surface displacement dataset, decadal velocity changes (Eq. 1) and velocity change uncertainties (Eq. 2) were computed using Top 50% average velocity over two periods: 2000–2014 (V1) and 2013–2024 (V2), across all PMAs. However, since relative velocity changes depend on the initial velocity magnitude (Eq. 1), velocity changes on PMAs with smaller magnitudes (<0.3 m yr⁻¹) exhibit higher uncertainties. According to our calculations, only 2% (n = 8) of the entire PMA dataset exhibits velocity changes greater than their respective uncertainties (σVchange;Fig. 8). Among these, 3 rock glaciers, 2 landslides, and 3 unclassified PMAs, were identified. These 3 rock glacier PMAs have an average size of 6,075 m² (~27 pixels) with a Top 50% average velocity of 0.59 m yr⁻¹. Two (one) of them, accelerate (decelerate) with a mean value of 198% (-46%). Landslide PMAs have an average size of 15,412 m² (~69 pixels) and a Top 50% average velocity of 2.5 m yr⁻¹. However, only 2 cases exhibit acceleration with a mean of 214%. PMAs in the 'unclassified' class have an average size of 7,050 m² (~31 pixels) and a Top 50% average velocity of 0.44 m yr⁻¹. One (two) accelerates (decelerates) with a mean value of 70% (-42%). […]*

[Figure]

*Figure 8. Modeling of relative velocity changes (dashed lines; Eq. 1) and their respective uncertainties (Eq. 2) for various velocity magnitudes (0.1 – 4 m yr⁻¹). Grey dots represent the entire PMA dataset. Blue, green and orange dots highlight PMAs where velocity changes exceed their uncertainties.*

**SC-8.** Assuming the errors in V1 and V2 are uncorrelated and their variances identical and equal to σ2V (none of these assumptions is invoked in the manuscript), ...

**Response >** The assumption of uncorrelated error is mentioned two times in the manuscript. The first one is mentioned in Section 4.4. Average spatial velocity and relative velocity changes in line 322:

> *[…] assuming that the NMAD for both periods is **similar and not correlated** (σV; cf. Section 5.4). Finally, […];*

*and more explicitly in Section 5.4. Reported uncertainties, In line 451:*

> *[…] The NMAD is **0.21 and 0.19** m yr⁻¹, for 2000-2014 and 2013-2024 periods, respectively. […];*

**SC-9.** Conversely, the authors' equation (2) replaces the numerator with V1 + V2. I mentioned last time that their equation can give meaningless negative results for negative values of V1 or V2.

**Response >** We acknowledge that there was an error in the derivation of Equation 1, which led to some misinterpretations. Equation 2 has now been corrected and is presented as follows:

$$\sigma V_{change} = \sigma V \frac{\sqrt{V_1^2 + V_2^2}}{V_1^2} \tag{2}$$

As mentioned in our previous response, since we are computing velocity changes relative to the first time period ($V_1$), negative values are expected. These negative values indicate deceleration between the two observation periods, which is a meaningful outcome in the context of slow-moving landslides and rock glaciers. Such landforms may experience deceleration or transition into a deactivation phase over decadal timescales, making negative velocity changes not only plausible but also physically consistent with known geomorphic behavior.

**SC-10.** b) Results: Section 5.5 The second half of the only paragraph states selected relative velocity changes, but no comprehensive presentation of quantitative estimates or their uncertainty is provided. While Fig. 5 is referenced, neither the relative velocity change nor its uncertainty is directly shown. Given the claims in the conclusion about the estimation of velocity changes, I feel the reader needs to be presented quantitative information to appraise the authors' claims.

**Response >** We have updated section 5.5 accordingly. Please refer to response of comment GC-1 in this manuscript to look at the modifications.

**SC-11.** Unclear basis for conclusions The conclusion sections continues to contain statement for which I cannot identify solid evidence.

In particular, a) A sentence I highlighted in the same context in the first round "Although underestimations occurred due to pixel size, temporal data gaps and velocity field heterogeneity, decadal velocity changes were observable under certain conditions, particularly for features exceeding 1 m a-1". What is the evidence that pixel size and temporal data gaps caused the observed underestimation? What is the precise meaning of observable, and how was this conclusion established? I assume you mean features whose velocity exceeds 1 m/a on average.

**Response >** The evidence supporting the underestimation of velocities is already presented in Section 5.3, and is illustrated in Figures 6 and 7, as well as in Table 2 (see the columns titled "Difference in velocity"). This issue is discussed as follows:

> *[…] Quantitatively, the average velocity differences between VHR and GNSS points is 0.01 ± 0.05 m yr-1 (Tapado complex) and 0.38 ± 0.3 m yr-1 (Largo rock glacier). Meanwhile, the average difference between L7/8 and GNSS points is 0.18 ± 0.24 m yr-1 (Tapado complex) and 1.35 ± 0.84 m yr-1 (Largo rock glacier; Figure 7). The good agreement on slow surface velocities on the Tapado complex could be explained by the homogeneous surface velocity field in both datasets (Fig. 6a). However, this consistency is not observed on the Largo rock glacier, where large differences are likely due to the heterogeneity of its surface velocity field. Figure 6c shows a single PMA that could be either divided in two, splitting Largo rock glacier in two different units, with likely independent dynamics. This is not the case for the VHR velocity field, showing rather a more homogeneous spatial distribution of velocities (Fig. 6d). […]*

The heterogeneous surface velocity fields derived from the L7/8 dataset over the Largo rock glacier (Figure 6) are primarily attributed to the size of the correlation windows used, along with the limited image texture and contrast. Although this rock glacier displays characteristic ridge-and-furrow geomorphology, the velocity patterns obtained from L7/8 imagery do not correspond well with the pseudo-ground control points (pseudo-GCPs) or with the surface velocity fields derived from the very high-resolution (VHR) dataset. These discrepancies are further addressed in Section 6.1.

*[…] In contrast, Largo rock glacier presents greater complexity. Despite its ridge-and-furrow morphology, its homogeneous texture (Fig. 2d) reduces contrast, potentially explaining observed discrepancies between the L7/8 and VHR results (3–4 m yr-1; Figure 6b). L7/8's lower resolution, which capture less surface detail, affects velocity estimates in landforms with high internal variability. Therefore, correlation parameters are key when performing image correlation (Kääb & Heid, 2012; Leprince et al., 2008; Rosu et al., 2015). As L7/8's smallest matching window (3x3 pixels, covering 2025 m²) differs substantially from the VHR window (7x7 pixels, covering 49 m²) leading to an averaging effect. This difference contributes to the observed variability in features such as Largo rock glacier. […]*

Based on these observations, we believe that sufficient evidence has been provided to support our interpretation. Furthermore, the validation of surface velocity fields using GNSS data reinforces the reliability of our interpretations and conclusions.

b) "Below this threshold, velocity changes detected with L7/8 data were not statistically significant." I am not aware of any formal statistical test that was conducted (see 1) and hence cannot identify the empirical basis for this claim

**Response >** We acknowledge that no formal statistical tests were performed in this study. Instead, we considered velocity changes to be significant only when they exceeded their respective uncertainty values. For further details regarding this approach and the modifications made, please refer to our response to comment GC-1. The corresponding sentence has been revised as follows:

> *[…] Although **some** underestimation occurred due to pixel size, temporal data gaps and velocity field heterogeneity, decadal velocity changes were **for 2% of PMA dataset (n = 8). Among these PMAs, we find acceleration (deceleration) in 2 (1) rock glaciers, 2 landslides, and 1 (2) unclassified PMAs, all exceeding their respective uncertainties. According to our calculations, detecting decadal velocity changes smaller than 0.4 m yr⁻¹ involves relatively high uncertainties when using L7/8 data.** […]*

**SC-12.** Presentation I encourage the authors, especially the senior authors, to make further improvements to the presentation. Language issues include missing articles (e.g. "Average velocity field" in l 296 [or plural]), incomplete sentences ("By considering PMAs with velocities exceeding 1 m a-1, nine rock glaciers and 2 landslides." in line 467); or sentences such as the following from line 312: "Stable areas were defined using TanDEM-X DEM and slopes lower than 35°, without taking into account neither glacier outlines with a buffer of 500 m for each glacier (RGI Consortium, 2017) nor all PMAs, also not confirmed ones produced in this study." Do you mean that glaciers and their surroundings (500 m buffer) and PMAs were excluded?

**Response >** We modify the text as suggested. Now you can read:

> *[…] over stable areas,defined using TanDEM-X DEM and slopes **below 35°. Glaciers outlines from RGI consortium (2017) and surroundings (with a 500 m buffer) and all PMAs—both confirmed and unconfirmed—were excluded**. Stable areas account for 53% of the study area (45x45 km²; Fig. S4). […]*

**SC-13.** This sentence is a representative example of the verbose style that I find difficult to follow. Adopting a more concise style and reducing the word count by ~20% could make this manuscript easier to follow.

**Response >** We agree with the reviewer that the writing style and wording required further improvement. Although the manuscript had already been reduced by 15% during the first revision, we conducted an additional thorough review, focusing on simplifying complex sentence structures—particularly those written in the past tense. This effort resulted in an additional 10% reduction in length, achieved without compromising clarity, coherence, or scientific content. We believe that further reduction would risk omitting essential information. In addition, language issues have been systematically addressed throughout the manuscript to ensure consistency and improve overall readability.

**SC-14.** Consider including a quantitative statement on the uncertainty estimate in abstract and conclusion

**Response >** We added an small assessment about the uncertainties in the abstract and the conclusions. Now you can read:

**Abstract:**

> *[…]* Nevertheless, decadal velocity changes were observed in 2% of PMAs, where two (one) rock glaciers show a significant acceleration (deceleration) over two decades. **Our calculations show that decadal velocity changes < 0.4 m yr$^{-1}$ are associated with high uncertainty when using L7/8 data, with sensitivity depending on the reference period.** Our results highlights *[…]*

**Conclusions:**

> *[…]* respective uncertainties. **According to our calculations, detecting decadal velocity changes below 0.4 m yr$^{-1}$ (two times decadal NMAD values) using L7/8 data involves high uncertainty, depending on both velocity magnitude and the length of the reference period.** The results of this study *[…]*

---

## Author Response (AR3)

**3rd round of review - Minor corrections**

**Response to comments by Handling editor Prof. Christian Hauck**

Dear authors,

Thank you very much for your revised version of the manuscript, which was sent out again in a second review round. Both reviewers are very satisfied with your changes, and there remain only minor comments to be addressed. From my point of view most of them can be addressed by going line per line through the manuscript again to detect typos and small inconsistencies as mentioned by the reviewer.

Please reply to these again point-by-point - we are looking forward very much to your revised manuscript and the completion of the review process.

Kind regards,
Christian Hauck
Editor

> Dear Prof. Hauck,
>
> Thank you very much for your message and for the opportunity to revise our manuscript once again. We are very grateful to you and the reviewers for the constructive feedback and the positive evaluation of our revised version.
>
> We have now carefully addressed all remaining comments and suggestions. As requested, we have gone through the manuscript line by line to correct typos and resolve minor inconsistencies. Please find below our detailed point-by-point response to the reviewers' final remarks, along with a clean and a tracked-changes version of the revised manuscript.
>
> We hope the revised version meets your expectations and remain at your disposal for any further clarification.
>
> Kind regards,
> Diego Cusicanqui, on behalf of all co-authors

**Response to comments by Dr. Jan Henrik Blöthe**

Below, I am commenting on the second round of revisions of the manuscript. I commented on the original submission and the first round of revisions before and I am very glad to see that the manuscript has significantly improved in quality and clarity. In their revised manuscript the authors have adjusted the text and figures largely in line with the comments made by the reviewers. The inclusion of additional figures, the more focused presentation of the uncertainties associated with the analysis as well as the condensation of the text has added to the scientific rigour of the work. That being said, there are still quite some typos remaining in the manuscript and some errors in the references to published work (e.g. Halla et al. 2020 is equal to Halla et al. 2021). I am recommending to critically go through the manuscript again and correct these. Additionally, I am outlining a few specific comments below (Line numbers refer to the document

"egusphere-2024-2393- manuscript-version3.pdf") that should in my view be addressed before accepting the manuscript for publication.

> Dear Dr. Blöthe,
>
> Thank you very much for your thorough and constructive feedback on our second revised version. We truly appreciate your positive evaluation regarding the improvements in clarity, scientific rigour, and presentation of uncertainties. We have carefully addressed all the specific points you raised. In particular, we have performed a meticulous line-by-line revision of the manuscript to eliminate remaining typos and inconsistencies.
>
> Please find below our detailed responses to your specific comments, as well as an updated version of the manuscript with tracked changes. We are grateful for your continued support in improving the quality of our work.
>
> Kind regards,
> Diego Cusicanqui, on behalf of all co-authors

there are still quite some typos remaining in the manuscript and some errors in the references to published work (e.g. Halla et al. 2020 is equal to Halla et al. 2021).

> Thank you for pointing out this error. We have carefully re-read the entire manuscript and checked all references. At least 50 typos were found and corrected.

Specific comments:

- L1: This might also just be a typo: The title should read "rock glacier kinematics" not "rock glaciers kinematics"

> Thanks for pointing this typo. We modified as suggested.

- L25-26: While I agree that this is an interesting finding, the reader might be confused why 2% of the data are put forward here. Isn't it more surprising (and therefore worth mentioning here) that 98% of the landforms show rather stable surface velocities?

> Our results show that velocity variations exceeding their respective uncertainties were observed in only 2% of the PMAs (n = 8). This means that for the remaining 98%, velocity changes could not be confidently detected inside the study period, as they remain within the uncertainty range. We have revised the manuscript text accordingly.
>
> *[...] However, decadal velocity changes **exceeding uncertainties** were observed **in only** 2% of PMAs, where two (one) rock glaciers **exhibit** significant acceleration (deceleration) over the past two decades. [...]*

> While the hypothesis of stable velocities across these PMAs is plausible, it is not the focus of the present study. We have however added a sentence in Section 6.4 (Andean velocity observations) to emphasize the stability of rock glacier velocities in the region including the latest paper published this year by [Blöthe et al., (2025)](). Now you can read:
>
> *[...] Such a level of acceleration might not be detected by L7/8 imagery, mainly due to the **high uncertainties (Fig. 8) and coarse spatial resolution (cf. Sec. 6.1). More recently, Blöthe et al. (2025) reported unchanged velocity change patterns on 175 rock glaciers over the past 50 years in the Valles Calchaquíes region (northwestern Argentina). Overall, our findings confirms limited rock glacier velocity changes in several regions of the Andes.** Further studies could benefit from incorporating older datasets, like SPOT 1-4 up to the mid 1980's or Corona images from the 1960s, **to extend temporal coverage and improve trend detection** (Dehecq et al., 2020; Kääb et al. 2021). [...]*

- L359: "show a linear trend in surface displacment" seems misleading here. The data shown in Fig. 5 show a linear trend in cumulated surface displacement, i.e. stable velocities.

> We modified the text as suggested. Now you can read:
> *[...] show a linear trend in **cumulative** surface displacement [...]*

- L362-363: I am surprised to see different numbers here than in the abstract and Table 1. Did I miss the explanation for the number of rock glaciers, landslides and unclassified being lower (rock glaciers, landslides) and higher (unclassified) here?

> We apologize for the error. These were typographical errors left over from the first version of the manuscript. We have changed it accordingly. Now you can read:
> *[...] PMAs (rock glacier = **153**; landslide = **124**; unclassified = **105**) [...]*

- L425-432: Like in the abstract, I think the 98% of data with stable surface displacement deserve to be mentioned here?

> We agree. We have added the following text:
> *[...] **Regarding the remaining 98% of PMAs (n = 374), velocity variations could not be confidently detected as they remain within the uncertainty range (Fig. 8; Table 2).** [...]*

- L513-514: Would be god to use the consistent terminology instead of "Top 50% pixels" here.

> Modified as suggested. Now you can read:
> *[...] When velocities are computed using the **Top 50% average velocity**, bias resulting from [...]*